# ON THE ROBUSTNESS OF DATASET INFERENCE

## ABSTRACT

Machine learning (ML) models are costly to train as they can require a significant amount of data, computational resources and technical expertise. Thus, they constitute valuable intellectual property that needs protection from adversaries wanting to steal them. *Ownership verification* techniques allow the victims of model stealing attacks to demonstrate that a suspect model was in fact stolen from theirs.

Although a number of ownership verification techniques based on watermarking or fingerprinting have been proposed, most of them fall short either in terms of security guarantees (well-equipped adversaries can evade verification) or computational cost. A fingerprinting technique introduced at ICLR '21, *Dataset Inference* (DI), has been shown to offer better robustness and efficiency than prior methods.

The authors of DI provided a correctness proof for linear (suspect) models. However, in a subspace of the same setting, we prove that DI suffers from high false positives (FPs) – it can incorrectly identify an independent model trained with non-overlapping data from the same distribution as stolen. We further prove that DI also triggers FPs in realistic, non-linear suspect models. We then confirm empirically that DI in the black-box setting leads to FPs, with high confidence.

Second, we show that DI also suffers from false negatives (FNs) – an adversary can fool DI by regularising a stolen model's decision boundaries using adversarial training, thereby leading to an FN. To this end, we demonstrate that black-box DI fails to identify a model adversarially trained from a stolen dataset – the setting where DI is the hardest to evade.

Finally, we discuss the implications of our findings, the viability of fingerprinting-based ownership verification in general, and suggest directions for future work.

## 1 INTRODUCTION

Machine learning (ML) models are being developed and deployed at an increasingly faster rate and in several application domains. For many companies, they are not just a part of the technological stack that offers an edge over the competitors but a core business offering. Hence, ML models constitute valuable intellectual property that needs to be protected.

Model stealing is considered one of the most serious attack vectors against ML models (Kumar et al., 2019). The goal of a model stealing attack is to obtain a functionally equivalent copy of a victim model that can be used, for example, to offer a competing service, or avoid having to pay for the use of the model.

In the *white-box* attack, the adversary obtains the exact copy of the victim model, for example by reverse engineering an application containing an embedded model (Deng et al., 2022). In contrast, in *black-box* attacks (known as *model extraction* attacks) (Papernot et al., 2017; Orekondy et al., 2019; Tramèr et al., 2016) the adversary gleans information about victim model via its predictive interface. Two possible approaches to defend against model extraction are 1) detection (Juuti et al., 2019; Atli et al., 2020; Zheng et al., 2022) and 2) prevention (Orekondy et al., 2020; Mazeika et al., 2022; Dziedzic et al., 2022). However, a powerful, yet realistic attacker can circumvent these defenses (Atli et al., 2020).

An alternative defense applicable to both white-box and black-box model theft is based on *deterrence*. It concedes that the model will eventually get stolen. Therefore, an *ownership verification* technique that can identify and demonstrate a suspect model as having been stolen can serve as a

deterrent against model theft. Early research in this field focused on *watermarking* based on embedding triggers or backdoors (Zhang et al., 2018; Uchida et al., 2017; Adi et al., 2018) into the weights of the model. Unfortunately, all watermarking schemes were shown to be brittle (Lukas et al., 2022) in that an attacker can successfully remove the watermark from a protected stolen model without incurring a substantial loss in model utility.

An alternative approach to ownership verification is *fingerprinting*. Instead of embedding a trigger or backdoor in the model, one can extract a fingerprint that matches only the victim model, and models derived from it. Fingerprinting works both against white-box and black-box attacks, and does not affect the performance of the model. Although several fingerprinting schemes have been proposed, some are not rigorously tested against model extraction (Cao et al., 2021; Pan et al., 2022) and others can be computationally expensive to derive (Lukas et al., 2021).

In this backdrop, *Dataset Inference* (DI), which appeared in ICLR 2021 (Maini et al., 2021) promises to be an effective fingerprinting mechanism. Intuitively, it leverages the fact that if model owners trained their models on *private data*, knowledge about that data can be used to identify all stolen models. DI was shown to be effective against white-box and black-box attacks and is efficient to compute (Maini et al., 2021). It was also shown not to conflict with any other defenses (Szyller & Asokan, 2022). Given its promise, the guarantees provided by DI merits closer examination.

In this work, we first show that DI suffers from false positives (FPs) — it can incorrectly identify an independent model trained with *non-overlapping data from the same distribution* as stolen. The authors of DI provided a correctness proof for a linear model. However, DI in fact suffers from **high FPs**, unless two assumptions hold: (1) a large noise dimension, as explained in the original paper and (2) a large proportion of the victim's training data is used during ownership verification, as we prove in this paper. Both of these assumptions are unrealistic in a subspace of the linear case used by DI: (i) we prove that large noise dimension can lead to low accuracy in the resulting model , and (ii) revealing too much of the victim's (private) training data is detrimental to privacy. Furthermore, we prove that DI also triggers FPs in realistic, non-linear models. We then confirm empirically that DI leads to FPs, with high confidence in the black-box verification setting, "*black-box* DI", where the DI verifier has access only to the inference interface of a suspect model, but not its internals .

We also show that black-box DI suffers from false negatives (FNs): an adversary who has in fact stolen a victim model can avoid detection by regularising their model with adversarial training. We provide empirical evidence that an adversary who steals the victim's dataset itself and adversarially trains a model can evade detection by DI.

We claim the following contributions:

- Following the same simplified theoretical analysis used by the original paper (Maini et al., 2021), in a subspace of the linear case used by DI, we show that for a linear suspect model, a) high-dimensional noise (as required in (Maini et al., 2021) leads to **low model accuracy** (Lemma 1, Section 3.1), and 2) DI **suffers from FPs** unless a large proportion of private data is revealed during ownership verification (Theorem 1, Section 3.1);

- Extending the analysis to non-linear suspect models, using a PAC-Bayesian framework (Neyshabur et al., 2018), we show that DI suffers from **FPs in non-linear models** regardless of how much private data is revealed (Theorem 2, Section 3.2.1);

- We empirically demonstrate the existence of **FPs in a realistic black-box DI** setting (Section 3.2.2);

- We show empirically that black-box DI also **suffers from FNs**: using adversarial training to regularise the decision boundaries of a stolen model can successfully evade detection by DI while incurring only a modest loss in accuracy ($\approx$ 6pp) (Section 4);

## 2 Dataset Inference Preliminaries

Dataset Inference (DI) aims to determine whether a *suspect model* $f_{SP}$ was obtained by an adversary $\mathcal{A}$ who has stolen a model ($f_\mathcal{A}$) derived from a victim $\mathcal{V}$'s private data $\mathcal{S}_V$, or belongs to an independent party $\mathcal{I}$ ($f_\mathcal{I}$). DI relies on the intuition that if a model is derived from $\mathcal{S}_V$, this information can be identified from all models. DI measures the *prediction margin*s of a suspect model around private and public samples: distance from the samples to the model's decision boundaries. If

Table 1: Summary of the notation used throughout this work.

| | | | |
|---|---|---|---|
| $\mathcal{V}$ | the victim | $f$ | a model |
| $\mathcal{I}$ | an independent party | $f_{\mathcal{V}}$ | a model trained on $\mathcal{S}_V$ |
| $\mathcal{A}$ | an adversary | $f_0$ | a model trained on $\mathcal{S}_0$ |
| $\mathcal{S}$ | a dataset | $f_{\mathcal{I}}$ | a model trained on $\mathcal{S}_I$ |
| $\mathcal{S}_V$ | $\mathcal{V}$'s private dataset | $f_{\mathcal{A}}$ | $\mathcal{A}$'s model |
| $\mathcal{S}_0$ | a public dataset | $f_{SP}$ | a suspect model |
| $\mathcal{S}_I$ | $\mathcal{I}$'s data | $\boldsymbol{w}$ | model weights |
| $\mathcal{D}$ | distribution that all datasets follow | $g_{\mathcal{V}}$ | regression model |
| $(\boldsymbol{x}, y)$ | a sample from $\mathcal{D}$ | $D$ | noise dimension |

$f_{SP}$ has distinguishable decision boundaries for private and public samples DI deems it to be *stolen*; otherwise the model is deemed *independent*.

In the rest of this section, we explain the theoretical framework that DI uses — consisting of a linear suspect model — the embedding generation necessary for using DI with realistic non-linear suspect models, and the verification procedure. A summary of the notation used throughout this work appears in Table 1.

## 2.1 THEORETICAL FRAMEWORK

The original DI paper (Maini et al., 2021) used a linear suspect model to theoretically prove the guarantees provided by DI. We first explain how DI works in this setting.

**Setup.** Consider a data distribution $\mathcal{D}$, such that any input-label pair $(\boldsymbol{x}, y)$ can be described as:

$$y \sim \{-1, +1\}, \boldsymbol{x_1} = y \cdot \boldsymbol{u} \in \mathbb{R}^K, \boldsymbol{x_2} \sim \mathcal{N}(0, \sigma^2 I) \in \mathbb{R}^D,$$

where $\boldsymbol{x} = (\boldsymbol{x_1}, \boldsymbol{x_2}) \in \mathbb{R}^{K+D}$ and $\boldsymbol{u} \in \mathbb{R}^K$ is a fixed vector. The last $D$ dimensions of $\boldsymbol{x}$ represent Gaussian noise (with variance $\sigma^2$).

**Structure of the linear model.** Assuming a linear model $f$, with weights $\boldsymbol{w} = (\boldsymbol{w_1}, \boldsymbol{w_2})$, such that $f(x) = \boldsymbol{w_1} \cdot \boldsymbol{x_1} + \boldsymbol{w_2} \cdot \boldsymbol{x_2}$, then the final classification decision is $sgn(f(x))$. With the weights initialized to zero, $f$ learns the weights using gradient descent with learning rate 1 until $yf(x)$ is maximized. Given a private training dataset $\mathcal{S}_V \sim \mathcal{D} = \{(x^{(i)}, y^{(i)})|i = 1, ..., m\}$, and a public dataset $\mathcal{S}_0 \sim \mathcal{D}$ (both of size $m$), then $\boldsymbol{w_1} = m\boldsymbol{u}$ and $\boldsymbol{w_2} = \sum_{i=1}^{m} y^{(i)} \boldsymbol{x_2}^{(i)}$ regardless of the batch size.

In DI, the prediction margin $p(\cdot)$ is used to imply the confidence of $f$ in its prediction. It is defined as the margin (distance) of a data point from the decision boundary.

$$p(x) \triangleq y \cdot f(x). \tag{1}$$

The authors (Maini et al., 2021) show that the difference of expected prediction margins of two datasets $\mathcal{S}_V$ and $\mathcal{S}_0$ is $D\sigma^2$. The threshold can be set $\lambda \in (0, D\sigma^2)$, and by estimating the difference of the prediction margins on $\mathcal{S}_0$ and $\mathcal{S}_V$ on $f_{SP}$, DI is able to distinguish whether that model is stolen.

Note that DI uses approximations of the prediction margins based on embeddings. The theoretical framework assumes that the approximations are accurate, and we can use them directly for the theoretical analysis (Equation 1). For the linear model, the margins can be computed analytically; however, in Section 2.2, we explain how the approximations of the margins are obtained.

## 2.2 EMBEDDING GENERATION

In order to use DI one needs to generate *embeddings* of the samples. $\mathcal{V}$ queries their model $f_{\mathcal{V}}$ with samples in their private dataset $\mathcal{S}_V$ and public dataset $\mathcal{S}_0$, and assigns the labels $b = 1$ and $b = 0$ respectively. The authors propose two methods of generating the embeddings: a white-box approach (MinGD) and a black-box one (Blind Walk). In this work, we use only Blind Walk as it outperforms

MinGD in most experimental setups in the original work, and is more realistic, as it only requires access to the API of the suspect model.

Blind Walk estimates the prediction margin of a sample by measuring its robustness to random noise. For a sample $(\boldsymbol{x}, y)$, to compute the margin, first choose a random direction $\delta$, and take $k \in \mathbb{N}$ steps in the same direction until the misclassification $f(\boldsymbol{x} + k\delta) \neq y$. This is repeated multiple times to increase the size of the embedding. As reported in (Maini et al., 2021), obtaining embeddings for 100 samples can take up to $30,000$ queries.

Having obtained the embeddings, $\mathcal{V}$ trains a regression model $g_\mathcal{V}$ that predicts the confidence that a sample contains private information from $\mathcal{S}_V$.

## 2.3 OWNERSHIP VERIFICATION

Using the scores from $g_\mathcal{V}$ and the membership labels, $\mathcal{V}$ creates vectors $\boldsymbol{c}$ and $\boldsymbol{c}_V$ of equal size from $\mathcal{S}_V$ and $\mathcal{S}_0$, respectively. Then for a null hypothesis $H_0 : \mu < \mu_V$ where $\mu = \bar{c}$ and $\bar{\mu} = \bar{c}_V$ are mean confidence scores. The test rejects $H_0$ and rules that the suspect model is 'stolen', or gives an inconclusive result.

To verify whether $f_{SP}$ is stolen or independent, $\mathcal{V}$ obtains the embeddings by querying the model (using Blind Walk) using samples from $\mathcal{S}_V$ and $\mathcal{S}_0$. Then they use the embeddings to obtain the confidence scores from the $g_\mathcal{V}$, and performs a hypothesis test on the two distributions of scores.

## 3 FALSE POSITIVES IN DATASET INFERENCE

To generate the embeddings for a specific sample in the private dataset $\mathcal{S}_V$, DI requires querying the suspect model $f_{SP}$ hundreds of times. To reduce the total number of queries, DI was shown to be effective with only 10 private samples with at least $95\%$ confidence. Additionally, DI requires a large random noise dimension $D$ such that probability of success increases to 1 as $D \to \infty$. In this section, we prove that these two assumptions are not realistic in the case of a linear model: 1) DI is susceptible to false positives (FPs) unless $\mathcal{V}$ reveals a large number of samples; 2) a large $D$ will harm the utility of the model (Section 3.1).

Furthermore, we find that the theoretical results on linear suspect models which say that the margins on different models are distinguishable with some strict conditions do not hold for more realistic non-linear suspect models. Using a PAC-Bayesian margin based generalization bound (Neyshabur et al., 2018) we prove that models trained on the same distribution are indistinguishable, and will trigger FPs (Section 3.2.1. Next, we provide empirical evidence for the existence of FPs (Section 3.2.2).

Figure 1: Probability of an FP as the fraction of revealed private samples for $D = 10$ for a linear suspect model (Equation 6). $\mathcal{V}$ needs to use many private samples to guarantee low false positive rate.

## 3.1 LINEAR SUSPECT MODELS

In section 2, we have a distribution $\mathcal{D}$ set up for linear models. The linear model $f$ should correctly classify most of the randomly picked data from this distribution. However, in a subspace of the linear case used by DI, we find that the dimension of the noise part of $\boldsymbol{x}$ needs to be small, otherwise it will harm the utility of the model.

**Lemma 1** (Need for Bounding Noise Dimension). *Let $f$ be a linear model trained on $\mathcal{S} \sim \mathcal{D}$. For a sample $(\boldsymbol{x}, y)$ sampled from $\mathcal{D}$ which is independent of $\mathcal{S}$, assuming that $\|\boldsymbol{u}\|_2 \leq \frac{1}{\sqrt{m}}$ and $\sigma^2 > \frac{1}{\sqrt{m}}$, then, the linear model $f$ correctly classifies $(\boldsymbol{x}, y)$ with a probability larger than $0.9$ only if $D < 10$.*

The details of the proof are in the Appendix A. Lemma 1 shows that if the dimension of $\boldsymbol{x}_2$, which follows $\mathcal{N}(0, \sigma^2)$, is large, then the noise will dominate $f$ and mislead it into making incorrect

predictions. For example, set $D = 1000$ and assume that the variance of $x_2$ is 0.25 (close to the CIFAR10 dataset). Then, $f$ can correctly classify a sample that is different from $f$'s training set with a probability up to 0.69.

**Theorem 1** (Existence of False Positives with Linear Suspect Models). *Let $f_\mathcal{I}$ be a linear classifier trained on the independent dataset $\mathcal{S}_I \sim \mathcal{D}$ with accuracy more than $0.9$. Assume that $|\mathcal{S}_I| = m$, $||u||_2 \leq \frac{1}{\sqrt{m}}$ and $\sigma^2 > \frac{1}{\sqrt{m}}$. Let $k$ be the number of samples estimated required for the verification. Then, the probability that $\mathcal{V}$ mistakenly decides that $f_\mathcal{I}$ is a stolen model $P[\Psi(f_\mathcal{I}, \mathcal{S}_V; \mathcal{D}) = 1] > 1 - \Phi(\frac{\sqrt{k}}{\sqrt{m}})$.*

Where $\Psi$ is $\mathcal{V}$'s decision function (Maini et al., 2021):

$$\Psi(f_{SP}, \mathcal{S}; \mathcal{D}) = \begin{cases} 1, \ if \ f_{SP} \sim f_\mathcal{A}, \\ 0, \ if \ f_{SP} \sim f_\mathcal{I}, \end{cases} \quad (2)$$

*Proof.* Recall that $\mathcal{V}$ tries to reveal only a few samples during the verification. For a distribution $\mathcal{D}$ where $||u|| \leq \frac{1}{\sqrt{m}}$ and $\sigma^2 > \frac{1}{\sqrt{m}}$.

Following the intuition from DI (Yeom et al., 2018), for satisfactory performance, DI must minimise both false positives and false negatives. Hence, the objective function is defined as:

$$min_\lambda \frac{\mathbb{P}[\Psi(f_\mathcal{I}, \mathcal{S}_V; \mathcal{D}) = 1] + \mathbb{P}[\Psi(f_\mathcal{V}, \mathcal{S}_V; \mathcal{D}) = 0]}{2}, \quad (3)$$

where the margin of $\mathcal{D}$ is estimated using $\mathcal{S}_V$ and $\mathcal{S}_0$. Note that we are only interested in the false positives $\mathbb{P}[\Psi(f_\mathcal{I}, \mathcal{S}_V; \mathcal{D}) = 1]$, let $\mathcal{S}_I = \{(x^{(i)}, y^{(i)}) | i = 1, ..., m\}$, $\mathcal{S}_*^k$ be a subset of $\mathcal{S}_*$ consisting of $k$ samples.

$$\begin{aligned} \mathbb{P}[\Psi(f_\mathcal{I}, \mathcal{S}_V; \mathcal{D}) = 1] &= \mathbb{P}[E_{(x,y) \in \mathcal{S}_V^k}[y f_\mathcal{I}(x)] - E_{(x,y) \in \mathcal{S}_0^k}[y f_\mathcal{I}(x)] \geq \lambda] \\ &= \mathbb{P}[E_{(x,y) \in \mathcal{S}_V^k}[\sum_i^m y^{(i)} x_2^{(i)} x_2] - E_{(x,y) \in \mathcal{S}_0^k}[\sum_i^m y^{(i)} x_2^{(i)} x_2] \geq \lambda] \\ &= \mathbb{P}[\frac{1}{k} \sum_j^k \sum_i^m y^{(i)} x_2^{(i)} x_2^{(j)} - \frac{1}{k} \sum_p^k \sum_i^m y^{(i)} x_2^{(i)} x_2^{(p)} \geq \lambda]. \end{aligned} \quad (4)$$

Recall that $x_2^{(i)}$, $x_2^{(j)}$ and $x_2^{(p)}$ are $D$-dimensional vectors sampled independently from $\mathcal{N}(0, \sigma^2)$. Using central limit theorem we can approximate the terms. We have $\sum_i^m y^{(i)} x_2^{(i)} \sim \mathcal{N}(0, m\sigma^2)$. Then, we can approximate $\frac{1}{k} \sum_j^k \sum_i^m y^{(i)} x_2^{(i)} x_2^{(j)}$ by $t_1 \sim \mathcal{N}(0, \frac{mD}{k}\sigma^4)$ and approximate $\frac{1}{k} \sum_p^k \sum_i^m y^{(i)} x_2^{(i)} x_2^{(p)}$ by $t_2 \sim \mathcal{N}(0, \frac{mD}{k}\sigma^4)$ (Maini et al., 2021). Thus, we get $t \sim \mathcal{N}(0, \frac{2mD}{k}\sigma^4)$, and

$$\mathbb{P}[\Psi(f_\mathcal{I}, \mathcal{S}_V; \mathcal{D}) = 1] = \mathbb{P}[t \geq \lambda] = \mathbb{P}[\sqrt{\frac{2mD}{k}}\sigma^2 Z \geq \lambda] = \mathbb{P}[Z \geq \frac{\sqrt{k}\lambda}{\sqrt{2mD}\sigma^2}] = 1 - \Phi(\frac{\sqrt{k}\lambda}{\sqrt{2mD}\sigma^2}), \quad (5)$$

where $Z \sim \mathcal{N}(0, 1)$. The optimal threshold is given as $\lambda = \frac{D\sigma^2}{2}$,

$$\mathbb{P}[\Psi(f_\mathcal{I}, \mathcal{S}_V; \mathcal{D}) = 1] = 1 - \Phi(\frac{\sqrt{kD}}{2\sqrt{2m}}). \quad (6)$$

From Equation 6, we see that the probability of false positives relies on the number of points used for the verification $\frac{k}{m}$ and the size of $D$. Combining with Lemma 1, the proof is complete. □

In other words, the success of DI is directly related to the number of samples used for the verification. This is similar to the analysis of failure of membership inference in the original paper when the $k$ is extremely low, e.g. only 10 samples. In the DI paper, it was explained that DI succeeds because it

calculates the average margin for multiple verification samples; whereas membership inference fails as it relies on per-sample decision. So when the number of tested samples is smaller, the success rate of DI will be close to 0.5, just like for membership inference. In Figure 1, we show the probability of an FP (Equation 6) for different values of $k$; even for $k = 10000$ the probability is 0.309 .

Hence, even the simple linear setup, $\Psi(f, \mathcal{S}; \mathcal{D})$ has false positives with high probability; in particular, when the fraction of tested samples is small.

### 3.2    Non-linear Suspect Models

Having demonstrated the limitations of the linear model, we now focus on non-linear suspect models. The intuition is based on the margin-based generalization bounds. Note that the generalization bounds states that the expected error of the margin based loss function is bounded, and the bound is mostly related to the distribution (Neyshabur et al., 2018). Since DI assumes all the datasets follow the distribution $\mathcal{D}$, our intuition is to directly use the generalization bounds and the triangle inequality to prove the similarity of the models trained on the same distribution.

#### 3.2.1    Theoretical Motivation

Let $f_{\boldsymbol{w}}$ be a real-valued classifier $f_{\boldsymbol{w}} : \mathcal{X} \to \mathbb{R}^k$, $||x|| \leq B$ with parameters $\boldsymbol{w} = \{W_i\}_{i=1}^d$. For any distribution $\mathcal{D}$ and margin $p(f, \boldsymbol{x}) = f(\boldsymbol{x})[y] - max_{j \neq y} f(\boldsymbol{x})[j] \leq \gamma$, where $\gamma > 0$. The margin is same as for the linear model with labels $y \in \{-1, +1\}$. Then, we define the margin loss function as:

$$\mathcal{L}_\gamma(f, y) = \mathbb{P}_{(\boldsymbol{x},y) \sim \mathcal{D}}[f(x)[y] - max_{j \neq y} f(x)[j] \leq \gamma]. \tag{7}$$

Note that the PAC-Bayes framework (Neyshabur et al., 2018) provides guarantees for any classifier $f$ trained on data from a given distribution. We define the expected loss of a classifier $f$ on distribution $\mathcal{D}$ as $\mathcal{L}_\mathcal{D} := E_{(\boldsymbol{x},y) \sim \mathcal{D}}[\mathcal{L}(f(\boldsymbol{x}), y)]$ and the empirical loss on a dataset $\mathcal{S}$ as $\hat{\mathcal{L}}_\mathcal{S} := \frac{1}{m} \sum_{(\boldsymbol{x},y) \in \mathcal{S}}[\mathcal{L}(f(\boldsymbol{x}), y)]$. Then, for a $d-$layer feed-forward network $f$ with parameters $\boldsymbol{w} = \{W_i\}_{i=1}^d$ and ReLU activation (Neyshabur et al., 2018). The empirical loss is very close to the expected loss. For any $\sigma, \gamma > 0$, with probability $1 - \sigma$ over the training set, we have:

$$|\mathcal{L}_\mathcal{D}(f_\mathcal{S}) - \hat{\mathcal{L}}_\mathcal{S}(f_\mathcal{S})| \leq \mathcal{O}(\epsilon), \tag{8}$$

where $\epsilon = \sqrt{\frac{B^2 d^2 h \ln(dh) \prod_{i=1}^d ||W_i||_2^2 \sum_{i=1}^d \frac{||W_i||_F^2}{||W_i||_2^2} + ln \frac{dm}{\sigma}}{\gamma^2 m}}$, and $h$ is the upper bound dimension for $\{W_i\}_{i=1}^d$.

This PAC-Bayes based generalization guarantee states that for a model $f$, the distance between the empirical loss and the expected loss is bounded, and the bound can be very small when the model's margin is large. Thus, we can expect that the margins of $f$ on any dataset that follows a given distribution to be similar. This contradicts the intuition of DI.

Moreover, since DI assumes that $\mathcal{S}_V$ and $\mathcal{S}_I$ follow the same distribution $\mathcal{D}$, we can show that the margins for $f_V$ and $f_\mathcal{I}$ are similar to each other.

**Theorem 2** (k-independent False Positives with Non-linear Suspect Models). *For the victim private dataset $\mathcal{S}_V \sim \mathcal{D}$ and an independent dataset $\mathcal{S}_I \sim \mathcal{D}$, let $f_{\boldsymbol{w}}$ be a $d-$layer feed-forward network with ReLU activations and parameters $\boldsymbol{w} = \{W_i\}_{i=1}^d$. Assume that $f_V$ is trained on $\mathcal{S}_V$ and $f_\mathcal{I}$ is trained on $\mathcal{S}_I$, $f_V$ and $f_\mathcal{I}$ have the same structure. Then, for any $B, d, h, \epsilon > 0$ and any $\boldsymbol{x} \in \mathcal{X}$, there exist a prior $\mathcal{P}$ on $\boldsymbol{w}$, s.t. with probability at least $\frac{1}{2}$,*

$$|E(p(f_V, \boldsymbol{x}) - p(f_\mathcal{I}, \boldsymbol{x}))| \leq \epsilon. \tag{9}$$

The details of the proof are in the Appendix A. Hence, for any two models trained on the same distribution, the expectation of margins for any sample are similar. Given that DI works by distinguishing the difference of margins for two models, it will result in false positives with probability at least $\frac{1}{2}$ (Theorem 2).

### 3.2.2 EMPIRICAL EVIDENCE

Having proved the existence of FPs for non-linear models, we now focus on empirically confirming it.

First, recall the original experiment setup (Maini et al., 2021); let us consider the following two models: 1) $f_{\mathcal{V}}$ trained using $\mathcal{S}_V$, and 2) $f_0$ trained using $\mathcal{S}_0$. In the original formulation, e.g. for CIFAR10, CIFAR10-train ($50,000$ samples) is used as $\mathcal{S}_V$, and CIFAR10-test is used as $\mathcal{S}_0$ ($10,000$ samples). Recall that $\mathcal{V}$ uses their $\mathcal{S}_V$ and $\mathcal{S}_0$ to obtain the embeddings that are then used to train the regression model $g_{\mathcal{V}}$.

DI was shown to be effective against several post-processing used to obtain *dependent models* which are expected to be flagged as stolen - true positives However, the independent model $f_0$ is trained on $\mathcal{S}_0$ — the same data that is used to train $g_{\mathcal{V}}$. This means that the same dataset $\mathcal{S}_0$ is used both to train $g_{\mathcal{V}}$ and subsequently, to evaluate it. This is likely to introduce a bias that overestimates the efficacy of $g_{\mathcal{V}}$ and DI as a whole.

To address this, and test whether DI works for a more reasonable data split, we use the following setup:

1) randomly split CIFAR10-train into two subsets ($A_{train}$ and $B_{train}$) of $25,000$ samples each;

2) assign $\mathcal{S}_V = A_{train}$, and train $f_{\mathcal{V}}$ using it;

3) continue using CIFAR10-test as $\mathcal{S}_0$ (nothing changes), and train $f_0$ using it;

4) $g_V$ is trained using the embedding for $\mathcal{S}_0$ and the new $\mathcal{S}_V$, obtained from the new $f_{\mathcal{V}}$;

5) assign $\mathcal{S}_I = B_{train}$, independent data of a third-party $\mathcal{I}$, who trains their model $f_{\mathcal{I}}$.

This way, we have an *independent* model $f_{\mathcal{I}}$ that was trained on data from the same distribution $\mathcal{D}$ as $\mathcal{S}_V$ but data that was not seen by $g_{\mathcal{V}}$[1].

Recall that to determine whether the model is stolen, DI obtains the embeddings for private ($\mathcal{S}_V$) and public ($\mathcal{S}_0$) samples. Then it measures the confidence for each of the embeddings using the regressor $g_{\mathcal{V}}$. For a model derived from $\mathcal{V}$'s $\mathcal{S}_V$, the mean difference ($\Delta\mu$) between the confidence assigned to $\mathcal{S}_V$ and $\mathcal{S}_0$ should be large. If the model is not derived from $\mathcal{S}_V$, the difference should be small. The decision is made using the hypothesis test that compares the distributions of measures from $g_{\mathcal{V}}$.

In Figure 2 we visualise the difference in the distributions for three models. For $f_{\mathcal{V}}$ we observe two separable distributions with a large ($\Delta\mu$), while for $f_0$ the difference is small — DI is working as intended. However, for $f_{\mathcal{I}}$, even though $\Delta\mu$ is smaller than for $f_{\mathcal{V}}$ it is sufficiently large to reject $H_0$ with high confidence. Therefore, $f_{\mathcal{I}}$ is marked as stolen, a false positive, In Table 2 we provide $\Delta\mu$ and the associated p-values for multiple random splits.

Table 2: Verification of an independent model trained on the same data distribution triggers an FP. Also, we report the accuracy of the models on the test set. We provide the mean and standard deviation computed across five runs. Verification done using $k = 10$ private samples. FPs become more significant as $k$ increases (see Appendix B).

| Model | Accuracy | $\Delta\mu$ | p-value |
|---|---|---|---|
| $f_{\mathcal{V}}$ | $0.87 \pm 0.03$ | $1.62 \pm 0.08$ | $10^{-18} \pm 10^{-18}$ |
| $f_{\mathcal{I}}$ | $0.87 \pm 0.03$ | $1.14 \pm 0.12$ | $10^{-8} \pm 10^{-8}$ |
| $f_0$ | $0.64 \pm 0.02$ | $-0.29 \pm 0.12$ | $0.46 \pm 0.04$ |

We discuss the implications of our findings in Section 5.

---

[1] We use the official implementation of DI, together with the architectures and training loops. Our changes are limited to the data splits only.

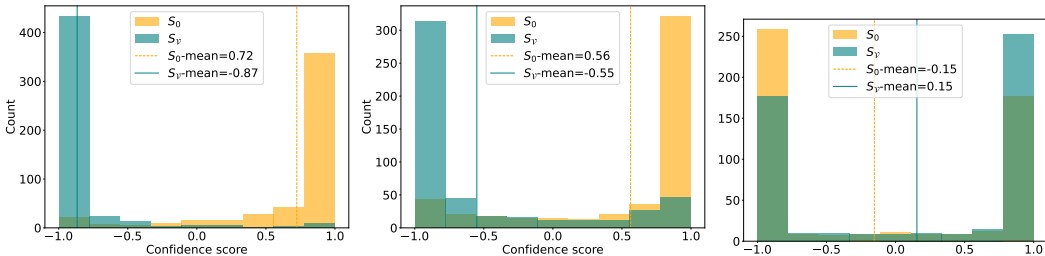

Figure 2: Left to right: $f_{\mathcal{V}}$, $f_{\mathcal{I}}$, $f_0$. Comparison of distributions of the confidence scores assigned to the embeddings by $g_{\mathcal{V}}$. $\Delta\mu$ is smaller for $f_{\mathcal{I}}$ than for $f_{\mathcal{V}}$ but large enough to trigger an FP.

Table 3: $f_{\mathcal{A}}$ adversarially trained on $\mathcal{S}_V$ results in a false negative. Also, we report the accuracy of the models on the test set. We provide the mean and standard deviation computed across five runs. Verification done using $k = 10$ private samples.

| Model | Accuracy | $\Delta\mu$ | p-value |
|---|---|---|---|
| $f_{\mathcal{V}}$ | $0.92 \pm 0.01$ | $1.59 \pm 0.04$ | $10^{-21} \pm 10^{-16}$ |
| $f_{\mathcal{A}}$ | $0.86 \pm 0.01$ | $0.12 \pm 0.06$ | $0.15 \pm 0.07$ |
| $f_0$ | $0.64 \pm 0.02$ | $-0.29 \pm 0.12$ | $0.46 \pm 0.04$ |

## 4 FALSE NEGATIVES IN DATASET INFERENCE

Having demonstrated the existence of false positives, we now show that DI can suffer from false negatives (FNs). $\mathcal{A}$ can avoid detection by regularising $f_{\mathcal{A}}$, and thus changing the prediction margins. This in turn, will mislead DI into flagging $f_{\mathcal{A}}$ as independent.

Recall that Blind Walk relies on finding the prediction margin by querying perturbed samples designed to cause a misclassification. In order to avoid detection, $\mathcal{A}$ needs to make the prediction margin robust to such perturbations. We do so using adversarial training: a popular regularisation method used to provide robustness against adversarial examples. $\mathcal{A}$ who launches a model extraction attack against $f_{\mathcal{V}}$, or steals $\mathcal{V}$'s $\mathcal{S}_V$ can adversarially train $f_{\mathcal{A}}$.

During adversarial training, each training sample $(\boldsymbol{x}, y)$ is replaced with an adversarial example that is misclassified $f_{\mathcal{A}}(\boldsymbol{x} + \gamma) \neq y$. There exist many techniques for crafting adversarial examples. We use *projected gradient descent* (Madry et al., 2018) (PGD), and we set $\gamma = 10/255$ (under $l_{\infty}$).

We evaluate adversarial training as a way to avoid detection in a setting where $\mathcal{A}$ steals $\mathcal{V}$'s $\mathcal{S}_V$ and trains their own model $f_{\mathcal{A}}$. $f_{\mathcal{A}}$ has the same architecture and hyperparameters as $f_{\mathcal{V}}$, but is adversarially trained. Hence, the experiment is biased in favour of DI[2].

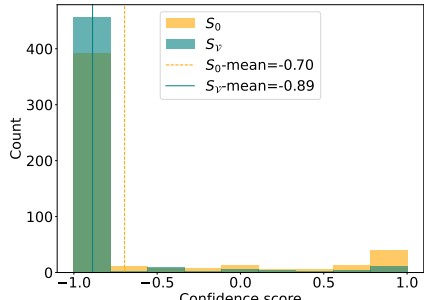

In Figure 3 we visualise the difference in the distributions of scores assigned by $g_{\mathcal{V}}$ to $f_{\mathcal{A}}$ embeddings derived for $\mathcal{S}_V$ and $\mathcal{S}_0$. We observe that the distributions are not clearly separable and result in low $\Delta\mu$, and hence $H_0$ cannot be rejected. Therefore, $f_{\mathcal{A}}$ is marked as an independent model, a false negative. In Table 3 we provide $\Delta\mu$ and the associated p-values for multiple runs.

Figure 3: Confidence scores assigned to embeddings by $g_{\mathcal{V}}$ obtained from $f_{\mathcal{A}}$. $\Delta\mu$ is small enough to trigger FNs.

Note that adversarial training comes with an accuracy trade-off. In our experiments, the accuracy of $f_{\mathcal{A}}$ goes from $0.92 \pm 0.01$ to $0.86 \pm 0.01$. We study how the amount of noise affects the verification in Appendix D. Also, we discuss the resulting implications in Section 5.

---

[2] We use the official implementation of DI, together with the architectures and training loops. Our changes are limited to adding adversarial training.

## 5 DISCUSSION

**Revealing private data.** We have shown that DI requires revealing significantly more than 50 samples to avoid false positives in the case of linear models (Figure 1). Since the core assumption of DI is that $S_V$ is private, revealing too much of $S_V$ during the ownership verification constitutes a privacy threat. In neither of the settings described in Section 5 of the original DI paper the victim *cannot query the model sufficiently* without leaking the query data to the adversary. Additionally, it was shown that using more samples gives $\mathcal{V}$ more information about the prediction margin than using stronger embedding methods (Maini et al., 2021). Model owners that operate in sensitive domains such as healthcare or insurance industry need to comply with strict data protection laws, and hence need to minimise the disclosure.

One potential way to protect the privacy of the private samples used for DI ownership verification is to use oblivious inference (Liu et al., 2017; Juvekar et al., 2018). This way $\mathcal{V}$ could query $f_{\mathcal{A}}$ without revealing $S_V$. Despite recent advances in efficient oblivious inference (Samragh et al., 2021; Watson et al., 2022; Samardzic et al., 2021; 2022), it requires *all* parties (including $\mathcal{A}$!) to update their software stacks which may not always be realistic.

**Viability of ownership verification using training data.** We have demonstrated that DI suffers from FPs when faced with an independent model trained on the same distribution. While it is reasonable to assume that $\mathcal{V}$'s data is private, the uniqueness of the distribution is difficult to guarantee in practice. For example, two model builders may have data from the same distribution because they purchased their training data from a vendor that generates per-client synthetic data from the same distribution (e.g., regional financial data). In fact, two model builders working on the same narrow domain and independently building models that are intended to represent the same phenomenon, may very well end up using data from the same distribution.

There are other methods that attempt to detect stolen models based on the dataset used to train them (Sablayrolles et al., 2020; Pan et al., 2022). However, they rely on flaws in the model to establish the ownership (susceptibility to adversarial examples (Sablayrolles et al., 2020) or membership inference attacks (Pan et al., 2022)). Intuitively, given a perfect membership inference attack, a fingerprinting scheme should be possible. However, recent work shows that for a balanced dataset, only a fraction of records is vulnerable to a confident membership inference attack (Carlini et al., 2022; Duddu et al., 2021) which in turn reduces the capabilities of a membership inference-based fingerprinting scheme. Therefore, any improvements to generalisation or robustness (such as adversarial training or purification (Nie et al., 2022)) of ML models reduce the surface for ownership verification schemes.

**White-box theft** Our experiments in Section 4 are limited to $\mathcal{A}$ that trains their own model — they either steal the data or conduct a model extraction attack. If $\mathcal{A}$ obtains an exact copy of the model, they might lack the data to fine-tune it with adversarial training. Hence, our findings do not apply to the white-box setting. We leave the examination of other threat models out as future work.

**Black-box vs. white-box verification setting**. Our evaluation is focused on the black-box DI setting. We do not consider the white-box DI setting which uses MinGD. While white-box DI is feasible in a scenario where $\mathcal{V}$ takes $\mathcal{A}$ (the holder of a suspect model) to court, requiring $\mathcal{A}$ to provide white-box access to the suspect model, prosecution is an expensive undertaking. Realistically $\mathcal{V}$ is likely to first conduct black-box DI to decide whether the expense of prosecution is justified. Therefore, FPs in the black-box DI setting can cause substantial monetary loss to $\mathcal{V}$.

## 6 CONCLUSION

We analyzed *Dataset Inference* (DI) (Maini et al., 2021), a promising fingerprinting scheme, to show theoretically and empirically that DI is prone to false positives in the case of independent models trained from distinct datasets drawn from the same distribution. This limits the applicability of DI only to settings where a model builder uses a dataset with a definitively unique distribution. We also showed that an attacker can use adversarial training to regularise the decision boundaries of a stolen model to evade detection by DI at the cost of a modest ($6pp$) drop in accuracy.

Nevertheless, DI is a promising ML fingerprinting scheme. Model owners can use our results to make informed decisions as to whether DI is appropriate for their particular settings.

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

## A  EXISTENCE OF FALSE POSITIVES IN DATASET INFERENCE

**Calculating the prediction margin.** We assume that the model weights are initialized to zero. For each sample $x$ in a dataset $\mathcal{S} \sim \mathcal{D} = \{(\boldsymbol{x^{(i)}}, y^{(i)})|i = 1, ..., m\}, y \sim \{-1, +1\}$. The learning algorithm observes all samples in $S$ once and maximize the loss function $L(\boldsymbol{x}, y) = y \cdot f(\boldsymbol{x})$. For the learning rate $\alpha = 1$, the weights are updates as:

$$w = w + \alpha y^{(i)} \boldsymbol{x^{(i)}}. \tag{10}$$

Recall that $\boldsymbol{x} = (\boldsymbol{x_1}, \boldsymbol{x_2}) \in \mathbb{R}^{K+D}$, the weights of the linear model are $\boldsymbol{w_1} = m\boldsymbol{u}$ and $\boldsymbol{w_2} = \sum_{i=1}^{m} y^{(i)} \boldsymbol{x_2^{(i)}}$ when the training is completed.

When writing out the linear classifier explicitly, we can easily calculate the prediction margin of each sample $(x, y)$ in $\mathcal{S}$,

$$y \cdot f(\boldsymbol{x}) = y \cdot (\boldsymbol{w_1 x_1} + \boldsymbol{w_2 x_2}) = y \cdot (m\boldsymbol{u} \cdot y\boldsymbol{u} + \sum_{i=1}^{m} y^{(i)} \boldsymbol{x_2^{(i)}} \cdot \boldsymbol{x_2}) = c + y \sum_{i=1}^{m} y^{(i)} \boldsymbol{x_2^{(i)}} \cdot \boldsymbol{x_2}. \tag{11}$$

The expectations of the prediction margin for the points in training set $\mathcal{S}^+ = \{(x, 1)|(x, 1) \in \mathcal{S}\}$ is,

$$\begin{aligned} E_{\mathcal{S}^+}[yf(\boldsymbol{x})] &= yc + E_{\mathcal{S}^+}[\sum_{i=1}^{m} y^{(i)} \boldsymbol{x_2^{(i)}} \cdot \boldsymbol{x_2}] = yc + E_{\mathcal{S}^+}[\sum_{\boldsymbol{x_2} \neq \boldsymbol{x_2^{(i)}}} y^{(i)} \boldsymbol{x_2^{(i)}} \cdot \boldsymbol{x_2} + y\boldsymbol{x_2^2}] \\ &= c + 0 + D\sigma^2. \end{aligned} \tag{12}$$

Note that in Equation 12, since $x_2 \sim N(0, D\sigma^2)$, then $x_2^2 \sim \chi^2$, $E[x_2^{(i)}] = D\sigma^2$.

Consider a new dataset $\mathcal{S}_0 \sim \mathcal{D}$, the expectations of the prediction margin for the points in $\mathcal{S}_0^+$ are,

$$E_{\mathcal{S}_0^+}[yf(x)] = yc + E_{\mathcal{S}_0^+}[\sum_{i=1}^{m} y^{(i)} x_2^{(i)} \cdot x_2] = c. \tag{13}$$

Finally, we see that the difference of prediction margin of training set $S$ and test set $\mathcal{S}_0$ is

$$E_{\mathcal{S}^+}[yf(x)] - E_{\mathcal{S}_0^+}[yf(x)] = D\sigma^2. \tag{14}$$

**DI's decision function.** From the above analysis, we know that the statistical difference between the distribution of training and test data is $D\sigma^2$ which is usually larger than 1 in numerical. DI utilizes this difference to predict if a potential adversary's model stole their knowledge.

Since we know that $E_{\mathcal{S}_0}[yf(x)] = c$ and $E_{\mathcal{S}}[yf(x)] = c + D\sigma^2$. Let $\Psi(f, \mathcal{S}; \mathcal{D})$ represent the dataset inference victim's decision function. It is defined as,

$$\Psi(f, \mathcal{S}; \mathcal{D}) = \begin{cases} 1, \ if \ E_{(x,y) \in \mathcal{S}}[y \cdot f(x)] - E_{\mathcal{D}}[y \cdot f(x)] \geq \lambda, \\ 0, \ otherwise, \end{cases} \tag{15}$$

where $\lambda \in [0, D\sigma^2]$ is some threshold that the decision function uses to maximise true positives and minimise false positives.

**Proof for Lemma 1** For a linear model $f$ trained on distribution $\mathcal{D}$ where $x = (x_1, x_2)$, $x_1 = yu$, $x_2 \sim \mathcal{N}(0, \sigma^2)$ and $||u||_2 \leq \frac{1}{\sqrt{m}}$, $f$ is expected to achieve high accuracy on any sample $(x, y)$ sampled randomly from $\mathcal{D}$ which is independent of the training data set of $f$.

*Proof.* Given a linear model $f$ trained on dataset $\mathcal{S} \sim \mathcal{D} = \{(x^{(i)}, y^{(i)}) | i = 1, ..., m\}$, and a test sample $(x, y)$ sampled randomly from $\mathcal{D}$ which is independent of $\mathcal{S}$, the probability that $(x, y)$ is correctly classified by $f$ can be represented as:

$$\mathbb{P}[yf(x) \geq 0] = \mathbb{P}[mu^2 + y \sum_{i}^{m} y(i) x_2^{(i)} x_2 \geq 0]$$

$$= \mathbb{P}[y \sum_{i}^{m} y(i) x_2^{(i)} x_2 \geq -mu^2] \tag{16}$$

$$\leq \mathbb{P}[y \sum_{i}^{m} y(i) x_2^{(i)} x_2 \geq -1]$$

Since $x_2 \sim \mathcal{N}(0, \sigma^2)$ are $D$-dimensional vectors, we can use central limit theorem to approximate the term. Thus, the internal term can be approximated by a variable $t \sim \mathcal{N}(0, mD\sigma^4)$. Let $Z \sim \mathcal{N}(0, 1)$,

$$\mathbb{P}[yf(x) \geq 0] \leq \mathbb{P}[\sqrt{mD}\sigma^2 Z \geq -1] = 1 - \Phi(-\frac{1}{\sqrt{mD}\sigma^2}) \tag{17}$$

where $\Phi$ is the normal CDF.

For a distribution where the randomness $\sigma^2 \geq \frac{1}{\sqrt{m}} \geq \frac{1}{4\sqrt{m}}$.

$$\mathbb{P}[yf(x) \geq 0] \leq 1 - \Phi(-\frac{4}{\sqrt{D}}), \tag{18}$$

where $\Phi(-\frac{4}{\sqrt{D}}) \approx 0.10$. The linear model $f$ can correctly classify a sample with a probability more than 0.9 only if $D < 10$.

We can also calculate the accuracy of the training set $\mathcal{S}$ similarly. For a training sample $(\boldsymbol{x}, y)$ sampled randomly from $\mathcal{S}$,

$$
\begin{aligned}
\mathbb{P}[yf(x) \geq 0] &= \mathbb{P}[m\boldsymbol{u}^2 + y \sum_i^m y(i)\boldsymbol{x_2}^{(i)} \boldsymbol{x_2} \geq 0] \\
&= \mathbb{P}[m\boldsymbol{u}^2 + y^2 \boldsymbol{x_2}^2 + y \sum_i^{m-1} y(i)\boldsymbol{x_2}^{(i)} \boldsymbol{x_2} \geq 0] \\
&= \mathbb{P}[y^2 \boldsymbol{x_2}^2 + y \sum_i^m y(i)\boldsymbol{x_2}^{(i)} \boldsymbol{x_2} \geq -m\boldsymbol{u}^2] \\
&\leq \mathbb{P}[y^2 \boldsymbol{x_2}^2 + y \sum_i^m y(i)\boldsymbol{x_2}^{(i)} \boldsymbol{x_2} \geq -1].
\end{aligned}
\tag{19}
$$

Since $y^2 \boldsymbol{x_2}^2 \geq 0$ for any sample in $\mathcal{S}$, we have

$$
y^2 \boldsymbol{x_2}^2 + y \sum_i^m y(i)\boldsymbol{x_2}^{(i)} \boldsymbol{x_2} \geq y \sum_i^m y(i)\boldsymbol{x_2}^{(i)} \boldsymbol{x_2}.
\tag{20}
$$

Then, $\mathbb{P}_{(\boldsymbol{x},y) \in \mathcal{S}} \geq \mathbb{P}_{(\boldsymbol{x},y) \in \mathcal{D}/\mathcal{S}}$. This completes the proof. $\qquad\square$

**Proof for Theorem 2** Let $f_{\boldsymbol{w}}$ be a $d$-layer feed-forward model trained on distribution $\mathcal{D}$ with parameters $\boldsymbol{w} = \{W_i\}_{i=1}^d$ and the ReLU activation function. Assuming a training dataset $\mathcal{S} \sim \mathcal{D}$, the model is given as $f_{\mathcal{S}} = f_{\boldsymbol{w}+\boldsymbol{u_S}}$, where $\boldsymbol{u_S}$ is a random variable whose distribution may also depend on $\mathcal{S}$.

Since the key to analyze the margin is the output of the model, we first introduce Lemma 2 that analyzes the perturbation bound of the model trained on $\mathcal{S}$ and $\mathcal{D}$.

**Lemma 2** (Perturbation Bound (Lemma 2) in (Neyshabur et al., 2018)). *For any $B, d > 0$, let $f_{\boldsymbol{w}} : \mathcal{X} \to \mathbb{R}^k$ be a $d-$layer neural network with ReLU activations. Then for any $\boldsymbol{w}$, and $\boldsymbol{x} \in \mathcal{X}$, and any perturbation $\boldsymbol{u_S} = \{U_i\}_{i=1}^d$ such that $||U_i||_2 \leq \frac{1}{d}||W_i||_2$, the change in the output of the network can be bounded as follow,*

$$
|f_{\boldsymbol{w}+\boldsymbol{u_S}}(\boldsymbol{x}) - f_{\boldsymbol{w}}(\boldsymbol{x})| \leq eB(\prod_{i=1}^d ||W_i||_2) \sum_{i=1}^d \frac{||U_i||_2}{||W_i||_2}.
\tag{21}
$$

Since our proof is also based on Lemma 2, it is analogous to the analysis of generalization bound in (Neyshabur et al., 2018) and is essentially the same for the first part.

*Proof.* The proof involves two parts. In the first part, we show the maximum allowed perturbation of parameters as shown in (Neyshabur et al., 2018). In the second part, we show that the margin difference of the models trained on $\mathcal{S}_V$ and $\mathcal{S}_I$ is also bounded by the perturbation of parameters. Let $\beta = (\prod_{i=1}^d ||W_i||_2)^{\frac{1}{d}}$, and consider a network with normalized weights $\tilde{W}_i = \frac{\beta}{||W_i||_2} W_i$. Due to the homogeneity of the ReLU, we have $f_{\tilde{\boldsymbol{w}}} = f_{\boldsymbol{w}}$. We can also verify that $(\prod_{i=1}^d ||W_i||_2) = \prod_{i=1}^d ||\tilde{W}_i||_2$ and $\frac{||W_i||_F}{||W_i||_2} = \frac{||\tilde{W}_i||_F}{||\tilde{W}_i||_2}$. Therefore, it is sufficient to prove the Theorem only for the normalized weights $\tilde{\boldsymbol{w}}$, and hence w.l.o.g we assume that for any layer $i$, $||W_i||_2 = \beta$.

Choose the distribution $\mathcal{P}$ of the prior of $\boldsymbol{w}$ to be $\mathcal{N}(0, \sigma^2 I)$, and consider the random perturbation $\boldsymbol{u_S} \sim \mathcal{N}(0, \sigma^2 I) = \{U_i\}_{i=1}^d$. Since the prior cannot depend on the learned model $\boldsymbol{w}$ or its norm, we set $\sigma$ based on the approximation $\tilde{\beta}$. For each value of $\tilde{\beta}$ on a pre-determined grid, we compute the PAC-Bayes bound, establishing the generalization guarantee for all $\boldsymbol{w}$ for which $|\tilde{\beta} - \beta| \leq \frac{1}{d}\beta$, and ensuring that each relevant value of $\beta$ is covered by some $\tilde{\beta}$ on the grid. We then take a union bound over all $\tilde{\beta}$ on the grid. For now, we consider a fixed $\tilde{\beta}$ and the $\boldsymbol{w}$ for which $|\beta - \tilde{\beta}| \leq \frac{1}{d}\beta$, and hence $\frac{1}{e}\beta^{d-1} \leq \tilde{\beta}^{d-1} \leq e\beta^{d-1}$.

Since $\boldsymbol{u}_{\mathcal{S}} \sim \mathcal{N}(0, \sigma^2 I)$, we get the following bound for the spectral norm of $U_i$ (Tropp, 2012):

$$\mathbb{P}_{U_i \sim \mathcal{N}(0,\sigma^2 I)}[||U_i||_2 > t] \leq 2he^{-t^2/2h\sigma^2}. \tag{22}$$

Taking a union bound over the layers, we get that with probability at least $\frac{1}{\sqrt{2}}$, the spectral norm of perturbation of $U_i$ in each layer is bounded by $\sigma\sqrt{2hln(2dh)}$. Plugging this spectral norm bound into Lemma 2 we have that with probability at least $\frac{1}{\sqrt{2}}$ the maximum allowed perturbation bound is:

$$max_{\boldsymbol{x}\in\mathcal{X}}|f_{\boldsymbol{w}+\boldsymbol{u}_{\mathcal{S}}}(\boldsymbol{x}) - f_{\boldsymbol{w}}(\boldsymbol{x})| \leq eB\beta^d \sum_i \frac{||U_i||_2}{\beta} \leq e^2 dB\tilde{\beta}^{d-1}\sigma\sqrt{2hln(2dh)} \leq \frac{\epsilon}{4}, \tag{23}$$

where $\sigma = \frac{\epsilon}{42dB\tilde{\beta}^{d-1}\sigma\sqrt{2hln(2dh)}}$. Then we can compute the difference of expectation margins for $f_{\mathcal{V}}$ which is trained on $\mathcal{S}_V$ and $f_{\mathcal{I}}$ which is trained on $\mathcal{S}_I$. Firstly, we compute the difference margins for any model $f_{\mathcal{S}}$ trained on $\mathcal{S} \sim \mathcal{D}$ and the target model $f_{\mathcal{D}}$. For any verified dataset $\hat{S} \in \mathcal{D}$,

$$|E(p(f_{\mathcal{S}}, \boldsymbol{x})) - E(p(f_{\mathcal{D}}, \boldsymbol{x}))|$$
$$=|E(f_{\boldsymbol{w}+\boldsymbol{u}_{\mathcal{S}}}(\boldsymbol{x})[y] - max_{j\neq y}f_{\boldsymbol{w}+\boldsymbol{u}_{\mathcal{S}}}(\boldsymbol{x})[j]) - E(f_{\boldsymbol{w}}(\boldsymbol{x})[y] - max_{j\neq y}f_{\boldsymbol{w}}(\boldsymbol{x})[j])|$$
$$=|(E(f_{\boldsymbol{w}+\boldsymbol{u}_{\mathcal{S}}}(\boldsymbol{x})[y]) - E(f_{\boldsymbol{w}}(\boldsymbol{x})[y])) - (E(max_{j\neq y}f_{\boldsymbol{w}+\boldsymbol{u}_{\mathcal{S}}}(\boldsymbol{x})[j]) - E(max_{j\neq y}f_{\boldsymbol{w}}(\boldsymbol{x})[j]))|$$
$$\leq max_{\boldsymbol{x}\in\mathcal{X}}(f_{\boldsymbol{w}+\boldsymbol{u}_{\mathcal{S}}}(\boldsymbol{x})[y] - f_{\boldsymbol{w}}(\boldsymbol{x})[y]) + max_{\boldsymbol{x}\in\mathcal{X}}(max_{j\neq y}f_{\boldsymbol{w}+\boldsymbol{u}_{\mathcal{S}}}(\boldsymbol{x})[j] - max_{j\neq y}f_{\boldsymbol{w}}(\boldsymbol{x})[j])$$
$$\leq 2max_{\boldsymbol{x}\in\mathcal{X}}|f_{\boldsymbol{w}+\boldsymbol{u}_{\mathcal{S}}}(\boldsymbol{x}) - f_{\boldsymbol{w}}(\boldsymbol{x})| \leq \frac{\epsilon}{2}. \tag{24}$$

So, for $f_{\mathcal{V}}$ trained on $\mathcal{S}_V$ and $f_{\mathcal{I}}$ trained on $\mathcal{S}_I$, we have with probability at least $\frac{1}{2}$ that the predictions margins are bounded by $\epsilon$:

$$|E(p(f_{\mathcal{V}}, \boldsymbol{x})) - E(p(f_{\mathcal{I}}, \boldsymbol{x}))|$$
$$\leq |E(p(f_{\mathcal{V}}, \boldsymbol{x})) - E(p(f_{\mathcal{D}}, \boldsymbol{x}))| + |E(p(f_{\mathcal{I}}, \boldsymbol{x})) - E(p(f_{\mathcal{D}}), \boldsymbol{x})| \tag{25}$$
$$\leq \epsilon.$$

$\square$

## B  IMPACT OF REVEALING MORE PRIVATE SAMPLES ON FALSE POSITIVES

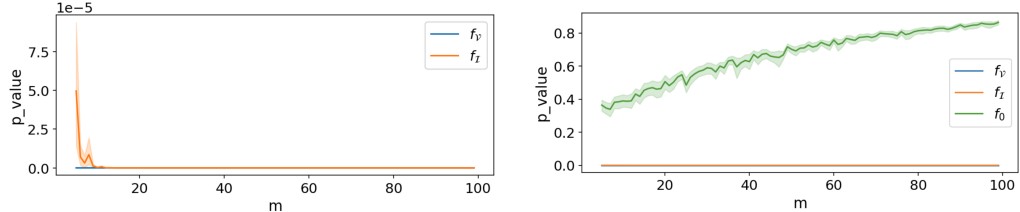

Figure 4: **Left:** Comparison of the verification confidence of $f_{\mathcal{V}}$ and $f_{\mathcal{I}}$. FP becomes stronger (lower p-value) as more samples are revealed. **Right:** same comparison, however, we include $f_0$ to show the desirable behaviour of an independent model.

In Figure 4, we show the results for verification, using Blind Walk, with more data (up to $k = 100$ private samples). As we increase the number of revealed private samples, the confidence of DI increases both for $f_{\mathcal{V}}$ (true positive) and $f_{\mathcal{I}}$ (false positive).

## C  RELATED WORK

**Model extraction detection and prevention.** Detection methods rely on the fact that many extraction attacks have querying patterns that are distinguishable from the benign ones (Juuti et al.,

2019; Atli et al., 2020; Zheng et al., 2022; Quiring et al., 2018). All of these can be circumvented by the adversary who has access to natural data from the same domain as the victim model (Atli et al., 2020). Prevention techniques aim to slow down the attack by injecting the noise into the prediction, designed to corrupt the training of the stolen model (Orekondy et al., 2020; Lee et al., 2019; Mazeika et al., 2022), or by making all clients participate in consensus-based cryptographic protocols (Dziedzic et al., 2022). Even though they increase the cost of the attack, they do not stop a determined attacker from stealing the model.

**Ownership verification.** There exist many watermarking schemes for neural networks (e.g. (Zhang et al., 2018; Uchida et al., 2017; Adi et al., 2018)) that have the same goal as DI does. However they were shown to be brittle (Lukas et al., 2022). It was shown that adversarial examples (Lukas et al., 2021) can be used to fingerprint a model or to watermark the dataset (Sablayrolles et al., 2020). However, adversarial training can be used to weaken both schemes (Lukas et al., 2021; Szyller & Asokan, 2022). On the other hand, if a model is sufficiently vulnerable to membership inference attacks, it can be used to fingerprint it (Pan et al., 2022).

## D  VERIFICATION WITH MORE NOISE

Table 4: Impact of the amount of noise (maximum number of perturbation steps) added during the verification on the success of DI (**baseline 50** steps). Using more noise does not prevent FNs against $f_\mathcal{A}$. However, it increases the standard deviation across all experiments, and has negative effect on the verification of $f_0$. We provide the mean and standard deviation computed over five runs. Verification done using $k = 10$ private samples. FNs highlighted in red.

| Model | Accuracy | Steps | $\Delta\mu$ | p-value |
|---|---|---|---|---|
| $f_\mathcal{V}$ | $0.92 \pm 0.01$ | **50** | $1.59 \pm 0.04$ | $10^{-21} \pm 10^{-16}$ |
| $f_\mathcal{A}$ | $0.86 \pm 0.01$ | 25 | $0.09 \pm 0.04$ | $0.09 \pm 0.07$ |
| | | **50** | $0.12 \pm 0.06$ | $0.15 \pm 0.07$ |
| | | 100 | $0.10 \pm 0.05$ | $0.08 \pm 0.09$ |
| | | 200 | $0.14 \pm 0.08$ | $0.16 \pm 0.11$ |
| $f_0$ | $0.64 \pm 0.02$ | **50** | $-0.29 \pm 0.12$ | $0.46 \pm 0.04$ |
| | | 100 | $-0.19 \pm 0.16$ | $0.37 \pm 0.12$ |

$\mathcal{V}$ who suspects that $\mathcal{A}$ might be using adversarial training to avoid the detection, can carry out the verification with more noise in order to escape the guarantees provided by adversarial training to $\mathcal{A}$.

In the experiments presented in Section 4, the average noise added during Blind Walk is $0.12 \pm 0.05$ (under $\ell_\infty$), and adversarial training is done with $\gamma = 10/255 (\approx 0.039)$. In this experiment, we vary the number of maximum steps taken by $\mathcal{V}$, and hence the maximum amount of noise added during the verification. We consider $\{25, 50, 100, 200\}$ steps (baseline 50 steps) which corresponds to $\{0.10 \pm 0.03, 0.12 \pm 0.05, 0.33 \pm 15, 0.38 \pm 23\}$ noise added (under $\ell_\infty$) during the verification.

Since $\mathcal{V}$ does not know which $f_{SP}$ is indeed stolen, in addition to $f_\mathcal{A}$, we also conduct this experiment for $f_0$ (for $\{50, 100\}$ steps).

In Table 4 we provide the results for the experiments with different amounts of noise. Using more steps does not improve the result against $f_\mathcal{A}$ compared to the baseline: 1) the standard deviation of the p-value increases; 2) we do not observe any linear relationship between the noise and $\Delta\mu$ or the associated p-value.

On the other hand, the confidence of the verification of $f_0$ decreases. The standard deviations of $\Delta\mu$ and its associated p-value increase. Nevertheless, the p-value remains sufficiently high.

In conclusion, increasing the amount of noise during Blind Walk does not allow $\mathcal{V}$ to circumvent $\mathcal{A}$'s adversarial training. Hence, DI remains susceptible to false negatives induced by adversarial training.

