# OpenReview forum: "On the Robustness of Dataset Inference"
_ICLR.cc/2023/Conference — Submitted to ICLR 2023_

### Official Review · Reviewer_ZZmg · 2022-10-17

**Confidence:** 5
**Correctness:** 2
**Technical Novelty And Significance:** 1
**Empirical Novelty And Significance:** Not applicable
**Recommendation:** 3

**Clarity, Quality, Novelty And Reproducibility:**

Clarity:
- There is room for improvement. Related work is very similar to the introduction. These two parts of the submission should be merged and the related work should be placed after the introduction - not at the end of the paper.
- In the introduction (3rd paragraph), a few defenses are mentioned from 2022 and then the authors claim that an attack from 2020 by Atli et al. can circumvent all of them. It looks as if it had been shown that the proposed attack in the paper from 2020 circumvented the defenses from 2022.
- In Figure 1, what is the linear suspect model 6? Is it from equation 6?

Neat:
- "many extraction attacks have querying patterns that are distinguishable from the benign ones queries Juuti et al. (2019);" remove ones or queries
- D, $g_v$ (distinguisher) should also be defined in Table 3.

Quality:
- The paper requires much more thorough experimentation.

Novelty:
- The novelty is limited since false positives and false negatives are expected for dataset inference.

Reproducibility:
- The authors did not submit the source code.

**Strength And Weaknesses:**

Strengths:

1. The authors provide a check of the defense technique proposed at ICLR 2021.
2. The paper provides both theoretical and experimental analysis.

Weaknesses:

1. Dataset inference is a statistical and not a deterministic method so the existence of FP (False Positives) and FN (False Negatives) is not surprising. The only important aspect is under which conditions such false alarms occur. Note that the ownership resolution is done through "statistical hypothesis testing, which takes the false positive rate $\alpha$ as a hyper-parameter and produces either conclusive positive results with an error of at most $\alpha$, or an 'inconclusive' result."
1. Regarding False Negatives - Adversarial training is more expensive thus stealing with this approach is not done. Additionally, adversarial training usually lowers the performance of a model (even in the experiments in this paper the authors report the drop in accuracy by 6pp) - which is why an adversary would not use it - the stolen service would be worse than the service exposed by competition - the victim. In Table 2, the accuracy values should be reported, as mentioned at the end of Section 4.
2. The assumption: "(2) a large proportion of the victim’s training data is used during ownership resolution/verification" can be fulfilled also by assuming a third-trusted party - an arbitrator - which is a realistic assumption for an ownership resolution (when considered as a court case). Note that revealing any data to an adversary might enable the adversary to retrain their model on the revealed data or simply claim ownership of the data. The approach with private or oblivious inference is also feasible, however, as pointed out - it is computationally intensive (hardware-based solutions might alleviate this overhead).
4. It is not shown how to obtain a false negative when a model is stolen from a private API by other means than decreasing the number of samples used for the ownership resolution. With respect to Section 5: "False negatives vs white-box theft.": What changes does an adversary have to make to fool dataset inference in the white-box setting when the victim model is simply copied by an attacker? For example, how should the adversary change the outputs from their stolen model to avoid detection by the dataset inference?
7. False negative is based only on the case where an adversary steals the private dataset but not when the adversary steals a model behind a publicly exposed API.
8. In Section 3.2.2 it should be also shown how dataset inference performs depending on the number of samples used for tests. The authors used the number of samples $k=10$, which is "extremely low", as stated at the end of Section 3.1 in this submission. A similar Figure to Figure 1 should be provided in Section 3.2.2. The authors showed a single/corner case instead of thoroughly analyzing the spectrum of possible cases for dataset inference. It is mentioned at the end of the caption for Table 1 that FPs become more significant as $k$ increases. This should be shown experimentally, at least in the appendix. The probability of an FP should be shown as the fraction (from 0 to 1) of revealed private samples (similarly to Figure 1).
9. Blind Walk and MinGD are two ways to estimate the prediction margin introduced in the dataset inference paper. For example, Tables 1 and 2 should be run for many more samples than $k=10$ and also with MinGD (not only with Blind Walk).
10. The assumptions for the theoretical results in Section 3 are different than those in the original paper. These should be stated explicitly.


**Summary Of The Paper:**

The paper analyzes the reactive defense against model stealing called dataset inference that was proposed at ICLR 2021. It is shown in this submission that dataset inference suffers from false positives (FP) and false negatives (FN). For FPs - it is presented that dataset inference can incorrectly resolve the model ownership when the independent model is trained with non-overlapping data from the same distribution as in the victim model. For FNs - it is claimed that dataset inference can be fooled when the stolen model is trained using adversarial training.

**Summary Of The Review:**

The paper requires more experiments, for example, to consider the whole spectrum of the number of data points used for dataset inference in the case of neural networks. The assumptions in the theoretical part are different than in the original work. Additionally, no suggestions are made that could improve the dataset inference defense.

---

> ### Author Response · Authors · 2022-11-10
> **Response 1**
>
> > 1. Dataset inference is a statistical and not a deterministic method so the existence of FP (False Positives) and FN (False Negatives) is not surprising. Note that the ownership resolution is done through "statistical hypothesis testing, which takes the false positive rate α as a hyper-parameter and produces either conclusive positive results with an error of at most α, or an 'inconclusive' result."
>
> In Section 3.1, we prove that in the linear case, the FP rate is around 0.5 when verifying with 50 samples. In Section 3.2, we prove that for a non-linear case, the FP rate is around 0.5 **no matter how many samples are used**.
>
> Recall that the hypothesis test is $H0:u<u_v$, where $u$ and $u_v$ are the prediction margins computed in the DI framework. The significant level $\alpha$ (called FPR in the original DI paper) means that assuming $u<u_v$ is true, then there will be at most $\alpha$ probability that this test outputs a wrong result saying that $u>u_v$.
> We show both theoretically and experimentally that an innocent model can have $u$ larger than $u_v$. Our results show that the p-value for an independent model is $10^{-8}$. We can set $\alpha = 10^{-7}$ and say that for any independent model, the prediction margin $u>u_v$. The probability for $u<u_v$ is $10^{-7}$.
>
> > 1a. The only important aspect is under which conditions such false alarms occur.
> From the theoretical results, in the linear case, to get FPR around 0.5, the condition is the number of verified samples is smaller than 50 (where the original DI paper says 50 is enough to ensure correctness of ownership verification). And in the non-linear case, there always exists $|u-u_v|<\epsilon, \forall \epsilon >0$ with probability 0.5.
>
> From the five experiments we conducted on different independent datasets, for an independent model, $u>u_v$ is correct with probability $1-10^{-7}$. Combined with our theoretical results, we think more than five different experiments are enough to show that the margins for different models have little relation to the decision of DI.
>
> > 2. Adversarial training is more expensive thus stealing with this approach is not done. Additionally, adversarial training usually lowers the performance of a model (even in the experiments in this paper the authors report the drop in accuracy by 6pp) - which is why an adversary would not use it - the stolen service would be worse than the service exposed by competition - the victim. In Table 2, the accuracy values should be reported, as mentioned at the end of Section 4.
>
> Prior work has suggested that adversarial training can be used to evade ownership verification schemes (see, e.g., [A])
>
> Robustness analysis of an ownership verification scheme needs to be based on a cost/benefit analysis from the attacker’s as well as victim’s perspective. One cannot declare a DI scheme as “secure” if the adversary will incur a minor loss in accuracy in defeating the ownership verification scheme! An adversary will certainly use a technique that incurs a small loss in accuracy if it will help the adversary evade detection. This is because now the adversary can offer a competing service at a much lower price. Potential customers may be willing to switch to using the adversary’s model even though its accuracy is slightly lower, if the cost to the customer is much lower. This is why you cannot disregard the small loss in accuracy we describe.
>
> → we will add accuracy values to Table 2.
>
> Also see our response to the Reviewer 1 (TtqN), question 3a.
>
> [A] - Deep Neural Network Fingerprinting by Conferrable Adversarial Examples, ICLR 2021, https://openreview.net/forum?id=VqzVhqxkjH1

---

> ### Author Response · Authors · 2022-11-10
> **Response 2**
>
> > 3. The assumption: "(2) a large proportion of the victim’s training data is used during ownership resolution/verification" can be fulfilled also by assuming a third-trusted party - an arbitrator - which is a realistic assumption for an ownership resolution (when considered as a court case). Note that revealing any data to an adversary might enable the adversary to retrain their model on the revealed data or simply claim ownership of the data. The approach with private or oblivious inference is also feasible, however, as pointed out - it is computationally intensive (hardware-based solutions might alleviate this overhead).
>
>
> First, the privacy concern has nothing to do with whether an arbitrator is involved or not! The concern is that each query to a suspect model leaks the victim’s sensitive data to the adversary (i.e. holder of the suspect model). Refer to the prologue of Section 5 of the original DI paper, where they clearly indicate that the adversary is assumed to be present during the ownership verification process.
>
> While oblivious inference can mitigate this problem in theory, it is not reasonable grounds to declare the problem solved exactly because, as you say, it is computationally intensive and hardware-based solutions incur further cost.
>
> We suggest that the reviewers consider the fact that we point out a shortcoming in the original paper **within the assumptions/desiderata made in that paper**. Whether there are mitigations based on additional assumptions (like hardware-assisted oblivious inference) that were **not** discussed in the original paper, published in ICLR, is beyond the scope of this review. That can be the topic for future work.
>
> > 4. It is not shown how to obtain a false negative when a model is stolen from a private API by other means than decreasing the number of samples used for the ownership resolution. With respect to Section 5: "False negatives vs white-box theft.": What changes does an adversary have to make to fool dataset inference in the white-box setting when the victim model is simply copied by an attacker? For example, how should the adversary change the outputs from their stolen model to avoid detection by the dataset inference?
>
> Could you clarify the following point: "It is not shown how to obtain a false negative when a model is stolen from a private API by other means than decreasing the number of samples used for the ownership resolution."
>
> We only show that false negatives in DI exist in a particular setting when the adversary steals the dataset and trains their model (which also holds for model extraction). We do not claim that DI is unsafe across all threat models. On the contrary, we specifically focus on two insights: a) the security guarantees of DI presented in the original work do not hold for all presented threat models; b) our work provides a more complete view of DI that a model builder can use to determine if DI is suitable for them. We leave other threat models out for future work.
>
> > 5. False negative is based only on the case where an adversary steals the private dataset but not when the adversary steals a model behind a publicly exposed API.
>
> We show that DI is susceptible to FNs in the stronger (for DI; see the original DI paper Table 1: $A_D$ vs $A_Q$) setting where the adversary steals the victim’s training data itself. An adversary who steals the model via model extraction can potentially induce FNs even more easily. We leave that as future work. Our goal in this paper is not to find all possible shortcomings of DI, but to demonstrate that, within the assumptions made by the original DI paper, it suffers from FPs and FNs in certain settings.
>
> > 6. In Section 3.2.2 it should be also shown how dataset inference performs depending on the number of samples used for tests. The authors used the number of samples k=10, which is "extremely low", as stated at the end of Section 3.1 in t. A similar Figure to Figure 1 should be provided in Section 3.2.2. The authors showed a single/corner case instead of thoroughly analyzing the spectrum of possible cases for dataset inference. It is mentioned at the end of the caption for Table 1 that FPs become more significant as k increases. This should be shown experimentally, at least in the appendix. The probability of an FP should be shown as the fraction (from 0 to 1) of revealed private samples (similarly to Figure 1).
>
> FPs become stronger with more queries.
>
> → We will add a plot similar to Figure 3 in the original DI paper. We provide such a figure below at : https://anonymous.4open.science/r/dirobustnessiclr2023-560D/fps_queries.png

---

> > ### Comment · Reviewer_ZZmg · 2022-11-21
> > **Oblivious Inference & Arbitrator**
> >
> > Regarding the private & oblivious inference, as discussed with reviewer PqLk: the references that you provided in the initial version of the submission were outdated and I fully agree that the computational cost can be decreased substantially with recent state-of-the-art methods and the *hardware-based solutions*, as also mentioned in my initial review.
> >
> > As mentioned in your answer to reviewer PqLk: faster oblivious inference can be achieved *by speeding up fully homomorphic encryption via hardware assistance*.
> >
> > >*First, the privacy concern has nothing to do with whether an arbitrator is involved or not!*
> >
> > If an arbitrator resolves the ownership then **no** training data points from the victim are revealed to the adversary.

---

> > > ### Author Response · Authors · 2022-12-08
> > > **Clarifications regarding oblivious reference**
> > >
> > > > Regarding the private & oblivious inference, as discussed with reviewer PqLk: the references that you provided in the initial version of the submission were outdated and I fully agree that the computational cost can be decreased substantially with recent state-of-the-art methods and the hardware-based solutions, as also mentioned in my initial review.
> > > As mentioned in your answer to reviewer PqLk: faster oblivious inference can be achieved by speeding up fully homomorphic encryption via hardware assistance.
> > >
> > > Your comment was posted after the discussion period had ended, and we were waiting for the AE to allow us to respond.
> > >
> > > Both in our response to the reviewer PqLk, and in the revised manuscript, we clarify that oblivious inference is not a realistic solution for black-box verification. While recent improvements reduce the computational overhead, other challenges remain - both parties need to update their stack to engage in the protocol. To avoid revealing private data, victim would require **every suspect** to do so.
> > >
> > > > If an arbitrator resolves the ownership then no training data points from the victim are revealed to the adversary.
> > >
> > > We emphasise that our focus is on the black-box case. Where the verifier (the victim or the arbitrator) carries out the black-box verification against the suspect's API. In such case, the training data points are still revealed to the suspect.
> > >
> > > A suspect who hands over their model will not observe any training data points. However, this is not required in the original DI paper. On the contrary, the original DI paper says that a black-box API access is sufficient to verify ownership using few queries (< 50 revealed training points).

---

> ### Author Response · Authors · 2022-11-10
> **Response 3**
>
> > 7. Blind Walk and MinGD are two ways to estimate the prediction margin introduced in the dataset inference paper. For example, Tables 1 and 2 should be run for many more samples than k=10 and also with MinGD (not only with Blind Walk).
>
> In this work, we focus on a black-box case (Blind Walk). MinGD requires white-box access by the victim/verifier to the model which is not realistic. The victim cannot get white-box access to every model that they suspect of theft.
> Additionally, as we point out in Section 2.2 (and so does the original DI paper in the Appendix C), Blind Walk outperforms MinGD.
>
> > 8. The assumptions for the theoretical in Section 3 are different than those in the original paper.
>
> The original DI paper assumes that on $u\in  \mathbb{R}^K$ and variance $\sigma \in \mathbb{R}^D$. We consider the case where $||u||_2\leq \frac{1}{\sqrt{m}}$ and $\sigma^2\geq\frac{1}{\sqrt{m}}$ where the variance of data is more pronounced.
> Consider this new set as $A=\{x|x\in\mathcal{D},x=[x1,x2],x1=yu\in \mathbb{R}^K,x2\sim N(0,\sigma^2 I) \in \mathbb{R}^D, ||u||_2\leq \frac{1}{\sqrt{m}}, \sigma^2\geq \frac{1}{\sqrt(m)} \}$ and the set DI paper uses as set $B = \{x|x\in\mathcal{D},x=[x1,x2],x1=yu\in \mathbb{R}^K,x2\sim N(0,\sigma^2) \in \mathbb{R}^D \}$, then we have $A\subseteq B$. These assumptions were stated clearly in both Lemma 1 and Theorem 1.

---

> > ### Comment · Reviewer_ZZmg · 2022-11-16
> > **MinGD**
> >
> > MinGD and BlindWalk methods were introduced in the original paper, and as stated in Section 5.1:
> >
> > **White-Box Setting:** *MinGD White-box embedding generation is used when $\mathcal{V}$ and $\mathcal{A}_∗$ resolve the claim for ownership in the presence of a neutral arbitrator, such as a court. Indeed, Kumar et al. (2020) highlight that such attacks potentially fall under Computer Fraud and Abuse Act in the USA and are prosecutable for reverse engineering the model’s ‘source code’.*
> >
> > Thus, MinGD should be analyzed as well (in this submission: Tables 1 and 2).

---

> > > ### Author Response · Authors · 2022-11-17
> > > **Response 3 contd.**
> > >
> > > Indeed both MinGD and Blind Walk were proposed in the original DI paper, and we did not claim otherwise. As we pointed out in the paper and our previous response, the original DI paper acknowledges that Blind Walk is a stronger method (original paper, Appendix C). Our empirical contributions in this paper is to point out gaps in the security claims of the original paper in the **black-box setting**. In our experiments, we make no claims about dataset inference in the white-box setting. The concerns we raise about the black-box setting are still relevant. For example, prosecution is expensive. A victim will likely first do black-box DI before deciding to take the risk of resorting to prosecution. Therefore FPs in the black-box setting can cause substantial monetary loss to the victim.
> > >
> > > Note also that the results of our theoretical analyses hold irrespective of the two methods.
> > >
> > > → we will clarify that (a) our focus is DI in the black-box setting, and (b) the above reasoning why FPs/FNs in the black-box setting are still important findings.

---

> > ### Comment · Reviewer_ZZmg · 2022-11-17
> > **Theorems 1 and 2**
> >
> > **Regarding Theorem 1:**
> >
> > Thank you for the clarification about the assumptions, I appreciate that. Based on Equation 6 which follows from the original proof in the Dataset inference paper, it is clear that the failure probability reduces with the reduction in the dimension size. After this step, the main contribution of this work is to set $D=10$ and claim that Dataset Inference fails. To achieve $D=10$, a contrived setup is created by setting extremely high variance in the data, and expecting the classifier to generalize well. Indeed, Dataset Inference fails when $D=10$, but it is also clear from the original theorem in the paper. The assumptions made to create a setup for enforcing $D=10$, regarding extremely high variance and very low mean are **unjustified**, and not of interest to practitioners who work in the over-parametrized regime.
> >
> > **Regarding Theorem 2:**
> >
> > Following the exchange with Reviewer 2 on the relevance of Theorem 2, I took a closer look at the applicability of this theorem in the deep model regime (as claimed by the authors). Unfortunately, I find multiple issues in the claims and expositions here as well. While most of the contents in section 3.2.1 and Appendix B are restated from the seminal work of Neyshabur et al. (2018). et. al., there are **key points that are overlooked**:
> > 1. Issue with Equation 8: Theorem 1 from Neyshabur et al. (2018) talks about the $\ell_0$ loss on unseen set (which is the 1-0 error) to be bounded in terms of the margin error on the training set, and a term that is dependent on the margin. This is not what Equation 8 says. It considers the same margin loss for both of them. This is incorrect. Please correct me if I am mistaken here. The conclusion that “This PAC-Bayes based generalization guarantee states that for a model $f$, the empirical loss is always close to the expected loss.” does not follow the same way when you correct for this. It is dependent on the margin.
> > 2. Theorem 2 states that the probability of prediction for a point by two different models is bounded by epsilon by claiming that this is true for values of $B,d,h,\epsilon > 0$. This hides the fact that $\epsilon$ is a dependent term, and makes the reader feel that $\epsilon$ can be set to be arbitrarily small. In fact, epsilon has a form that depends on the variance, the dimension, the bound $B$, and many other terms which are also provided after Equation 3 in the original paper, and restated after Equation 23 in your work. In the absence of this context, the value of $\epsilon$ in itself is meaningless and the bound is uninformative.

---

> > > ### Author Response · Authors · 2022-11-18
> > > **Response 3 contd2**
> > >
> > > > Based on Equation 6 which follows from the original proof in the Dataset inference paper, it is clear that the failure probability reduces with the reduction in the dimension size. After this step, the main contribution of this work is to set D=10 and claim that Dataset Inference fails.
> > >
> > > In Theorem 1,  we prove that in our linear case, when the dimension is limitted, the probability of FP relies on the number of verified samples. This properity holds whenever the noise dimension has a small upper bound. D=10 is just a representative number to show our results, we can also use D<100 to show this properity.
> > >
> > > > To achieve D=10, a contrived setup is created by setting extremely high variance in the data, and expecting the classifier to generalize well. Indeed, Dataset Inference fails when D=10, but it is also clear from the original theorem in the paper. The assumptions made to create a setup for enforcing D=10, regarding extremely high variance and very low mean are unjustified, and not of interest to practitioners who work in the over-parametrized regime.
> > >
> > > The high variance considered in our setup is common, because in practice the useless features could be larger  than the useful features. For example, consider a dataset with gender and age features to predict the gender. In this case, the age is the noise feature, but the variance of age feature can be much larger than the variance of gender feature.
> > >
> > > Moreover, it is enough to show a proof is wrong by showing a counter-example. The counter-example is important -- this importance is proved by what the original DI paper have done-- they extend the linear counter-example proposed in [1] to a larger space and prove their correctness successfully.
> > >
> > > $\rightarrow$ We will make the subspace relation (counter-example) clearly in our paper.
> > >
> > > > Issue with Equation 8: Theorem 1 from Neyshabur et al. (2018) talks about the ℓ0 loss on unseen set (which is the 1-0 error) to be bounded in terms of the margin error on the training set, and a term that is dependent on the margin. This is not what Equation 8 says. It considers the same margin loss for both of them. This is incorrect. Please correct me if I am mistaken here. The conclusion that “This PAC-Bayes based generalization guarantee states that for a model f, the empirical loss is always close to the expected loss.” does not follow the same way when you correct for this. It is dependent on the margin.
> > > Our statement is consistent with Neyshabur et al.(2018).
> > >
> > > We can use a simple reduction $|L_r(D)-\hat{L_r(S)}| \leq |L_0(D)-L_r(D)|+|L_0(D)-\hat{L_r(S)}|$, where $L_r(D) = \hat{L_r(D)}$ and use their theorem twice to get the results we use since $\sigma$ is arbitrarily chosen.
> > >
> > > ->we will change the conclusion to "the distance between the empirical loss and the expected loss is bounded, and the bound can be very small when the model's margin is large".
> > >
> > > > Theorem 2 states that the probability of prediction for a point by two different models is bounded by epsilon by claiming that this is true for values of B,d,h,ϵ>0. This hides the fact that ϵ is a dependent term, and makes the reader feel that ϵ can be set to be arbitrarily small. In fact, epsilon has a form that depends on the variance, the dimension, the bound B, and many other terms which are also provided after Equation 3 in the original paper, and restated after Equation 23 in your work. In the absence of this context, the value of ϵ in itself is meaningless and the bound is uninformative
> > >
> > > $\epsilon$ is not a dependent term. $\sigma$ is a dependent term which is dependent on $\epsilon$. So $\epsilon$ can be any constant larger than 0. Since $\sigma$ is a dependent term, we use the word "there exist" in our Theorem.
> > >
> > > [1]Nagarajan V, Kolter J Z. Uniform convergence may be unable to explain generalization in deep learning[J]. Advances in Neural Information Processing Systems, 2019, 32.

---

### Official Review · Reviewer_PqLk · 2022-10-21

**Confidence:** 5
**Correctness:** 3
**Technical Novelty And Significance:** 4
**Empirical Novelty And Significance:** 4
**Recommendation:** 8

**Clarity, Quality, Novelty And Reproducibility:**

__Clarity__: The paper is very well written, and the authors' thought process is easy to follow, making the paper very easy to follow and read.

__Quality and Novelty__: This work tackles a very practical concern regarding the practicality and downsides of dataset inference, which has been proposed (and has also had follow-up work) as an ownership resolution method. The theoretical analyses are insightful and highlight the problem with DI, vis-a-vis false positives and false negatives. Not all aspects of the research are "novel" per se (like the technique for false negatives) but overall I think it is impactful work.

__Reproducibility__: Standard datasets and models are used, along with fairly simple computations, but it would be nice to include an implementation (with exact seeds) for perfect reproducibility.

### Minor Comments

- Please fix the citation style. In most places, references that should be of the form .... (Author et. al.) are instead ... Author et. al, with the citation part of the text itself, making it confusing in places while reading.
- Section 1. """...via its inference interface" -> "...via its predictive interface". Prediction is not the same as inference. (see [this](https://www.datascienceblog.net/post/commentary/inference-vs-prediction/) blog for a good explanation).
- Above Theorem 2: "Thus, we can expect that the.....to be similar". This statement seems far-fetched: similar losses do not automatically imply similar decision boundaries.
- Dataset Inference has also been extended (with improvements) to the case of unsupervised learning [1]. It is a relatively new paper (so okay to not have included it in the initial submission) but might be worth looking at.
- Section 4: "...and is trained the same way as $f_\mathcal{V}$" do both use adversarial training?
- Section 5: "...constitutes a privacy threat": I disagree. The central authority involved in the ownership resolution process is trusted (which is why it does the resolution in the first place, and models are released to it after all), and __can__ be trusted to not misuse this data.
- Section 5: "....that can cause significant overhead." - how much? Given the conflict in ownership, the overhead just might be worth it.
- Table 3 (notations) should be in the main paper if the authors are able to find the space to fit it.
- In (18), please introduce $\Phi$ before using it

#### References

[1] Dziedzic, Adam, Haonan Duan, Muhammad Ahmad Kaleem, Nikita Dhawan, Jonas Guan, Yannis Cattan, Franziska Boenisch, and Nicolas Papernot. "Dataset Inference for Self-Supervised Models." arXiv preprint arXiv:2209.09024 (2022).

**Strength And Weaknesses:**

## Strengths

- Discussion on False Positives starting with the theoretical analyses drives the point home, showing that the flaw is critical to the approach itself, and not just an empirical loophole explored by the adversary. All the proofs are detailed, thorough, and easy to follow.

- Analyses are thorough, with proper visuals and numbers (with error margins, to highlight variance and stability of results), as well as realistically-sized models and datasets. All the figures are very helpful in understanding the points being made with their aid.

- The inclusion of False Negatives is very useful and much needed; any adversary that is malicious enough to steal data would not stop from going a step further and evading such detection techniques. It definitely opens up the sub-field to the same kind of cat-and-mouse game that adversarial machine learning is subject to, which isn't necessarily a bad thing (and can be good in fact).

## Weaknesses

- I'm not convinced about the applicability of Theorem 2. It states that False Positives occur with P at least 0.5, but this is true when _one_ sample is used for DI, right? Assuming $n$ samples are used, wouldn't the expected FPR bound decay with $2^{-n}$?

- Section 4: to account for potentially perturbed inputs, can the party executing Blind Walk not simply adjust to a higher step size (perturbation)? At least some form of adaptive defense should be evaluated for a complete picture.

- I have some concerns with some of the reductions, especially in Appendix B. They can probably be fixed by showing more intermediate steps, but would like some clarification from the authors:

  - In (11), $c=m\cdot u\cdot y^2$ ; since this has a $y$ term, how can it be treated as a constant?
  - In (12), $c$ from (11) becomes $y\cdot y$ (which makes me wonder if (11) has a typo), and the $y$ term outside the summation (present in (11)) seems to be missing here.
  - "Note that in Equation 12..." here assumes that each $y_i=1$, which may not be true- should the term here not be $D\sigma^2\sum_{i} y_i$ instead?
  - In (13), the same $yc$ term is used, but would it really be the same?
  - Assuming $\Phi$ in (18) refers to the Normal CDF: 0.9 here seems like an arbitrary choice. Also, is $\Psi(-\frac{1}{\sqrt{10}})=0.38$? Not sure where the 10 or 0.9 appears from this.

**Summary Of The Paper:**

This work does a deep dive into dataset inference, an up-and-coming ownership verification technique for machine learning models. The authors show how DI suffers from a nontrivial FPR in both theoretical and practical settings and FNRs in practical settings with simple adversarial modifications. These analyses highlight how DI can falsely accuse trainers of stealing data, while at the same time failing to detect actual stealing adversaries.

**Summary Of The Review:**

The paper makes some very good arguments about the validity and utility of dataset inference, with both theoretical analyses and empirical experiments that bring to light the existence of false positive and false negative cases, which can be harmful in practical scenarios. Apart from a few minor comments and some hiccups in the proofs (which I think might be a mix of some typos and unexplained intermediate steps, so nothing that can't be fixed), I think it would be a good addition to ICLR.

---

> ### Author Response · Authors · 2022-11-10
> **Response 1**
>
> > 1. I'm not convinced about the applicability of Theorem 2. It states that False Positives occur with P at least 0.5, but this is true when one sample is used for DI, right? Assuming n samples are used, wouldn't the expected FPR bound decay with 2−n?
>
> Yes, the result presented in Theorem 2 is for one sample. And the probability is $2^{-n}$ if we want to make every sample become a false positive. But we do not need every sample to be a false positive, we need the expectation of margins to be a false positive.
> We get the result from Lemma 2, which provides the upper bound for any sample in the distribution as $|f_{w+u_s}(x) - f_{w}(x)|\leq Bound$. This bound should still hold for the average of the dataset $\frac{\sum_i^m|f_{w+u_s}(x) - f_{w}(x)|}{m}\leq Bound$. Therefore we can get the expectation of  prediction margin with all samples (m) which is the same as the current result.
>  →We will make Theorem 2 suitable for the expectation of prediction margins: $E(p(f_V,\{x})-p(f_I, {x}))\leq \epsilon$.
>
> > 2. Section 4: to account for potentially perturbed inputs, can the party executing Blind Walk not simply adjust to a higher step size (perturbation)? At least some form of adaptive defense should be evaluated for a complete picture.
>
> During adversarial training, the perturbation is bounded to eps=10/255 (~0.039) under linf. During Blind Walk, the amount of noise added to the sample is $0.12\pm0.05$ (under linf). In other words, the amount of noise used during the verification is already, on average, three times as large as that used during training.
>
> > 3. I have some concerns with some of the reductions, especially in Appendix B. They can probably be fixed by showing more intermediate steps, but would like some clarification from the authors:
>
> We assume the binary classification case as we mentioned in Section 2 where $y\sim\{+1,-1\}$.
>
> →We will clarify this in the Appendix.
>
> We use the ‘positive’ part $S^{+}$ of the training data where $S^{+}$ contains all the training samples where the label is +1. The ‘negative’ part is the same as the positive part so we omit it.
> > In (11), $c = muy^2$; since this has a $y$ term, how can it be treated as a constant?
>
> Since $y\in\{+1,-1\}$, we have $y^2  = 1$. $c$ is a constant where $c=m{u}^2$.
> > In(12), $c$ from (11) becomes $yy$(which makes me wonder if (11) has a typo), and the $y$ term outside the summation seems to be missing here.
>
> We construct a set $S^{+}$ where all samples have a label +1. Our reductions after (11) all use this ‘positive’ set. So $E_{S^{+}}(yf({x})) = E_{S^{+}}(yyf({x})) = yc + yy[...] $. Since $y^2=1$, the $y$ outside the summation equals 1 and we omit it.
>
>  > “Note that in Equation (12)…” here assumes that each $y^i = 1$, which may not be true-should the term here not be $D\sigma^2\sum_y y^i$ instead?
>
> You are right, it is a typo. The $\sum_{i=1}y^i(x_2^i)^2$ should be $\sum_{x_2^i = x_2}y^i (x_2^i)^2 = yx_2^2$.  We mean to separate the summation in equation (11) into two parts, $x_2^i=x_2$ and $x_2^i \neq x_2$.
> Now the summation in equation(12) becomes $ \sum_{x_2 \neq x_2^i}y^i x_2^i x_2 + \sum_{x_2^i=x_2} y^i (x_2^i)^2$. The expectation of the first term is 0. And in the second term, since $x_2^i=x$ and $(x,y) \in S^{+}$, in this term $y^i=y=1$.
>
> > In (13), the same $yc$ term is used, but would it really be the same?
>
> Since ${u}$ is fixed for distribution $\mathcal{D}$ where $S^{+},S_{0}^{+} \sim \mathcal{D}$, and $m$ is the same for both $S^{+}$ and $S_{0}^{+}$, these two equations  have the same $yc$ where $y=1$.
>
> For the ‘negative’ $S^{-}$ case where the labels ($y$ and $y^i$ of the $x_2^i=x$) of the samples are -1. $-E(yf(x)) = E(yyf(x)) = yc + yy [\sum y^i…] = -c + E[\sum_{x_2^i \neq x_2}y^i x_2^i x_2  + \sum_{x_2^i = x_2} y^i (x_2^i)^2 ]   = -c - D\sigma^2$. So $E(y(fx)) = c+ D\sigma^2$ which is the same with the positive case. We omit this negative set case because the reductions are the same with the positive set $S^{+}$.
>
> > Assuming $\Phi$ in (18) refers to Normal CDF: 0.9 here seems like an arbitrary choice. Also, is $\Phi(\frac{-1}{\sqrt(10)}) = 0.38$? Not sure where the 10 or 0.9 appears from this.
>
> For (18), since $\sigma^2\geq\frac{1}{\sqrt{m}}\geq\frac{1}{4\sqrt{m}}$, we use $\Phi(\frac{-4}{\sqrt(10)})=0.10$ to get 0.9 and D=10.
>
> → We will add this to our proof.

---

> > ### Comment · Reviewer_PqLk · 2022-11-11
> > **Thanks for the Clarifications**
> >
> > 1. Thanks for clearing that up, but I am still confused about the statement. It shows that for any B, d, h, $\epsilon$, x, there exists some prior $\mathcal{P}$ on $w$, but it doesn't say anything about the prior being the same for all $x$ considered, right? Maybe when looking at some $x_1$ the prior would be different from some other point $x_2$, which would then stop any average-based reduction (as outlined in your response above).
> >
> > 2. Yes, but that doesn't mean that higher magnitudes cannot be tried. The idea was to increase this noise to account for the change in the overall environment, and not just have a relative difference in noise (which is anyway large). I still believe the authors should include results for a larger step size (if time permits, of course).
> >
> > 3. Thanks for the clarifications. I had missed the {$+1, -1$} bit, and it makes sense now. Glad to see the typo (and the consequent equation) has been fixed.
> >
> > 4. I am still confused about the result from (18). $\frac{1}{\sqrt{M}} > \frac{1}{4\sqrt{M}}$ seems arbitrary here. Why the 4 (and even 10, for that matter)? Also, please add a clarification that this is indeed the Normal CDF (should not have to assume).

---

> > > ### Author Response · Authors · 2022-11-11
> > > **Response 1 contd.**
> > >
> > > > 1. Thanks for clearing that up, but I am still confused about the statement. It shows that for any B, d, h, ϵ, x, there exists some prior P on w, but it doesn't say anything about the prior being the same for all x considered, right? Maybe when looking at some x1 the prior would be different from some other point x2, which would then stop any average-based reduction (as outlined in your response above).
> > >
> > > Since $f_w: \mathcal{X}->\mathbb{R}^k$ where $w$ is the weights of $f_w$, The prior is for the weights, so the bound holds for the whole data space $x\in\mathcal{X}$.
> > >
> > > > 2. Yes, but that doesn't mean that higher magnitudes cannot be tried. The idea was to increase this noise to account for the change in the overall environment, and not just have a relative difference in noise (which is anyway large). I still believe the authors should include results for a larger step size (if time permits, of course).
> > >
> > > → We will start these experiments now. Since it needs multiple iterations over multiple models, we may not be able to complete the experiments before the discussion period ends. We will add the results to the next revision of the paper once the experiments are completed.
> > >
> > > > 4. I am still confused about the result from (18). $\frac{1}{\sqrt{M}} > \frac{1}{4\sqrt{M}}$ seems arbitrary here. Why the 4 (and even 10, for that matter)? Also, please add a clarification that this is indeed the Normal CDF (should not have to assume).
> > >
> > > It is chosen to get the probability 0.9. Compared to the inequality, we want to use a specific number to make this lemma (and following Theorem) easier to understand. Having a specific number makes the following analysis easier to visualise.
> > >
> > > → We will add this clarification.

---

> ### Author Response · Authors · 2022-11-10
> **Response 2**
>
> > 4. Minor:
>
> → We will address the stylistic comnets.
>
> > Above Theorem 2: "Thus, we can expect that the.....to be similar". This statement seems far-fetched: similar losses do not automatically imply similar decision boundaries.
>
> Note that in the PAC-Bayes based framework, the prediction margin is defined as the loss function. So, this margin loss can be directly related to the margin we want, which is different from the traditional loss functions like CE, MSE and so on.
>
> > Section 4: "...and is trained the same way as  fV " do both use adversarial training?
>
> Only $f_A$ is trained with adversarial training.
> → We’ll make the phrasing clearer.
>
> > Section 5: "...constitutes a privacy threat": I disagree. The central authority involved in the ownership resolution process is trusted (which is why it does the resolution in the first place, and models are released to it after all), and can be trusted to not misuse this data.
>
> In a black-box ownership verification setting, the entity holding the suspect model (a potential adversary) **sees all queries** sent to it during the ownership verification process. This is the source for the privacy concern. It does not matter if there is a trusted central authority involved in the process.
>
> The original DI paper recognizes this threat. This is why they want to minimise the number of the revealed private samples (see the original DI paper Section 5.2: “It is important for the victim to resolve ownership claims in as few queries as possible, since each query involves the victim revealing part of their private dataset $S_V$“). The original DI paper limits the number of queries to 50. This is important because the premise of the DI paper is that training datasets are private.
>
> Also, see our responses to Reviewer 1 (TtqN) question 1a, and Reviewer 3 (Zzmg) question 3.
>
> > Section 5: "....that can cause significant overhead." - how much? Given the conflict in ownership, the overhead just might be worth it.
>
> Prior work on oblivious inference reported overheads of 1000x or more (e.g., [B,C,D]). Furthermore, use of oblivious inference requires changes to the software stacks of both verifiers as well as all potential suspect models.
>
> [B] - CryptoNets: applying neural networks to encrypted data with high throughput and accuracy, ICML 2016, https://dl.acm.org/doi/10.5555/3045390.3045413
>
> [C] - Oblivious Neural Network Predictions via MiniONN Transformations, CCS 2017, https://dl.acm.org/doi/10.1145/3133956.3134056
>
> [D] - GAZELLE: A Low Latency Framework for Secure Neural Network Inference, USENIX 2018, https://www.usenix.org/conference/usenixsecurity18/presentation/juvekar

---

> > ### Comment · Reviewer_PqLk · 2022-11-11
> > **Thanks for the response**
> >
> > I'd like to thank the authors for clarifying my doubts and responding to my queries. I'm glad to see that the authors have a plan for the revision, and look forward to seeing it!
> >
> > Regarding oblivious inference: these references are quite dated. The current state-of-the-art is much faster, with very realistic runtimes[1] and support for running with minimal software changes. Some of these methods are even fast enough to _train_ entire VGG networks in under a day [2]. Getting around with no OT is obviously going to be faster, but the authors should not completely dismiss these mechanisms based on results that are 4-6 years old. The fact that this was not mentioned in the original paper does not mean it cannot be discussed in subsequent works that discuss it. Acknowledging limitations is crucial, and comparisons should be fair (as opposed to using outdated benchmarks to show alternatives being 1000x slower).
> >
> > ### References
> >
> > [1] Samragh, Mohammad, et al. "On the application of binary neural networks in oblivious inference." Proceedings of the IEEE/CVF Conference on Computer Vision and Pattern Recognition. 2021.
> >
> > [2] Watson, Jean-Luc, Sameer Wagh, and Raluca Ada Popa. "Piranha: A {GPU} Platform for Secure Computation." 31st USENIX Security Symposium (USENIX Security 22). 2022.

---

> > > ### Author Response · Authors · 2022-11-11
> > > **Response 2 contd.**
> > >
> > > Thank you for pointing out the references to faster MPC protocols;
> > > → we will add them to the paper as well as two other references we found which claim fast oblivious inference by speeding up fully homomorphic encryption via hardware assistance [E, F]. We agree with you that the computational overhead can be brought down to more reasonable numbers with these recent advances. Although computational overhead can be brought down, other challenges with deploying oblivious inference for DI remain – all parties need to update their software stack, including the entity holding the suspect model requiring a setting where the prover and verifier co-operate (which was not a requirement for the original DI).
> > >
> > > → We will update the discussion about oblivious inference to reflect the above.
> > >
> > > Also, increasing the number of queries helps reduce FPs only in the theoretical analysis with a simple linear model and that in the experiments increasing queries did not bring down the FP rate (see Table 1)
> > >
> > > → In the Appendix, we will also include a figure that illustrates the results for the verification with more samples: https://anonymous.4open.science/r/dirobustnessiclr2023-560D/fps_queries.png
> > >
> > > [E] - F1: A Fast and Programmable Accelerator for Fully Homomorphic Encryption, MICRO 2022, https://dl.acm.org/doi/10.1145/3466752.3480070
> > >
> > > [F] - CraterLake: a hardware accelerator for efficient unbounded computation on encrypted data, ISCA 2022, https://dl.acm.org/doi/10.1145/3470496.3527393

---

> > > > ### Comment · Reviewer_PqLk · 2022-11-12
> > > > **Thanks**
> > > >
> > > > I would like to thank the authors for their clarifications, and engaging in conversation despite my already 'accept' rating (where most authors would have considered responding as a low priority).
> > > >
> > > > My concerns have been addressed, and I agree that a note on oblivious transfer should be added to the revised version.I have no further questions- best of luck!

---

### Official Review · Reviewer_TtqN · 2022-10-25

**Confidence:** 4
**Clarity, Quality, Novelty And Reproducibility:** I included the evaluation above.
**Correctness:** 2
**Technical Novelty And Significance:** 2
**Empirical Novelty And Significance:** 2
**Recommendation:** 5

**Details Of Ethics Concerns:**

No concern.

**Strength And Weaknesses:**

Strengths:

1. This paper suggests a false sense of security that the dataset inference mechanism may make.
2. The paper empirically shows some downsides of the dataset inference mechanisms.

Weaknesses:

1. It's a bit unclear why reducing the number of queries is important.
2. It's also unclear why the existence of false positives is the weakness.
3. It's further unclear why the attacker performs adversarial training to construct a model.
4. The paper has large room for improving the quality of the writing and the presentation.

Detailed comments:

I like the research question that this paper asks: "what would be the failure modes of the dataset inference?" which is the highlighted work in the last year's ICLR. The paper approaches this question theoretically and empirically, and it shows in the evaluation that there are such cases.

However, the major problem of this paper is that the failure modes that this paper exposes do not seem to be the actual weakness of the dataset inference. Moreover, some scenarios, such as an adversary training a model with adversarial training (for evasion), are misleading. Thus, I believe the paper is not ready to appear in ICLR 2023.

Here, I provide my detailed comments on the weaknesses.


[False Positives with Smaller Number of Queries]

If I understood the threat model correctly, the attacker trains a model with the stolen dataset, and the victim wants to identify whether the attacker uses the stolen data. In this case, the model trained by the adversary is already open to the victim; thus, I am a bit confused why we assume a smaller number of queries bounds the victim. Of course, reducing the number of required queries is good for efficacy. However, as long as the victim can query the model sufficiently, the dataset inference seems to work.

More generally, I imagine a scenario where two parties are at the court, and one party has to show that the other party's model uses the data extracted from the first party's model. In this case, unless the verification process takes days or years, we could allow the victim to do as many queries as possible. Running 50k queries on CIFAR10 only takes a few minutes (at most).


[Existence of False Positives]

I am a bit skeptical about the problem of false positives. Just having some false positives is insufficient to claim that the dataset inference is unsafe. I would expect some more strong evidence or claims, such as "even if the victim uses 50k queries, the false positive is still around 50%." It is, then, indeed a problem.

I like the results showing that if the number of private samples is large, the false positives become higher. But on second thought, it's somewhat trivial as the victim has to make predictions on many private samples for the ownership claim. Moreover, there is no reason why the victim wants to use a larger number of private samples. I would imagine if the victim uses a smaller number of private samples, then correctly estimating the membership of a subset of the entirety could be sufficient to claim the ownership.


[Use of Adversarial Training as a Defense]

The paper shows that an adversary can increase false negatives by training their model using adversarial training. This is confusing because I could not imagine a scenario where the adversary harms the accuracy of their model by running adversarial training. One of the purposes of stealing private data is to achieve state-of-the-art accuracy; then, running adversarial training contradicts the motivation.

I also assume that many trivial things could achieve the same as running adversarial training. One example would be running with DP-SGD. As the dataset inference is the membership identification, training a model with DP can reduce the inference success. BUT, the private model trained by the adversary cannot perform well; even for CIFAR10, it achieves ~75% accuracy with the epsilon 7-8.


[Minors]

1. The citation is not compatible with the ICLR; "Deng et al. (2022)" is better "[Deng et al., 2022]."
2. Many notations appear over reading; better to summarize by creating the "Notations" section.
3. Sec 2 largely depends on the setup proposed by Maini et al. and omits many details; for example, "what are the embeddings?" or "what is the white box (MinGD)?" It would be nice to re-write the entire section so that readers without prior knowledge of the dataset inference can understand the details clearly.
4. It is a bit difficult to understand the paragraph starting with "The authors ..."; it should be re-written.
5. The introduction contains sufficient related work, so I wonder if the paper needs more in Sec 6.



**Summary Of The Paper:**

This paper analyzes the dataset inference mechanism from an offensive perspective. The paper starts with the theoretical analysis of the failure scenarios, such as the existence of false positives. The paper analyzes linear cases and then extends it to non-linear cases. The paper empirically demonstrates that there are indeed false positives when the victim uses a smaller number of queries. The paper also shows some potential cases of false negatives when the attacker employs adversarial training. The paper concludes with the possible limitations of the dataset inference.


**Summary Of The Review:**

The paper studies the weaknesses of the dataset inference mechanism. I agree that exposing a protection mechanism's weaknesses is essential to understand its possible risks. However, the weaknesses this paper shows seem not to be the actual weaknesses of the dataset inference. I can safely imagine that those weaknesses are not any concern in practical scenarios. Thus, I am leaning a bit more toward rejecting this paper.

---

> ### Author Response · Authors · 2022-11-10
> **Response 1**
>
> > 1. Unclear why reducing the number of queries is important [False Positives with Smaller Number of Queries].
>
> > If I understood the threat model correctly, the attacker trains a model with the stolen dataset, and the victim wants to identify whether the attacker uses the stolen data. In this case, the model trained by the adversary is already open to the victim; thus, I am a bit confused why we assume a smaller number of queries bounds the victim. Of course, reducing the number of required queries is good for efficacy. However, as long as the victim can query the model sufficiently, the dataset inference seems to work.
>
> We consider a black-box setting for ownership verification: the adversary obtains a model (either via model extraction or by stealing the data) and e.g., launches a competing service. The victim, or a verifier acting on behalf of the victim, has only black-box (i.e. query) access to that suspect model. The victim/verifier cannot freely inspect its internals.
> In DI, the victim/verifier queries the suspect model using records from its **private** training data. The query is necessarily revealed to the entity holding the suspect model. The key assumption in the DI paper (see the original DI paper Section 5.2: “It is important for the victim to resolve ownership claims in as few queries as possible, since each query involves the victim revealing part of their private dataset $S_V$“) is that the victim wants to minimise the number of queries so as not to reveal their private dataset to external parties. The original paper limits the number of queries to 50 in its experiments. This is important because the premise of the DI paper is that datasets are private.
> Note that in neither of the settings described in Section 5 of the original DI paper the victim **cannot** “query the model sufficiently” without leaking the query data to the adversary.
> → This explanation is already in Section 5. We will try to clarify it better.
>
> Additionally, in our empirical evaluation with non-linear models, the use of more queries **does not reduce the probability of an FP** but makes it worse: we mention this in Table 1.
> → we will add experimental results with more data to the Appendix. We provide such a figure below: https://anonymous.4open.science/r/dirobustnessiclr2023-560D/fps_queries.png

---

> ### Author Response · Authors · 2022-11-10
> **Response 2**
>
> > 2. Unclear why the existence of false positives is the weakness [Existence of False Positives].
>
> > 2a. Just having some false positives is insufficient to claim that the dataset inference is unsafe. I would expect some more strong evidence or claims, such as "even if the victim uses 50k queries, the false positive is still around 50%." It is, then, indeed a problem.
>
> In all our experiments, **all** independent models trained on the same distribution trigger false positives (Section 3.2.2).
>
> Minimizing queries for reasons of dataset privacy (see answer to #1 above) is a desirable property as identified in the original DI paper. Note that we do **not** claim that “dataset inference is unsafe”. We are careful to characterize our results in the paper as follows: (a) the security properties of DI deviate from what is claimed in the original paper in certain settings (Section 3 and 4), and (b) our work can inform model builders to make informed decisions about whether DI is safe to use _in_their_setting_ (Section 7). We suggest that this is a valuable addition to understanding the limits and applicability of DI, which, as we point out in Section 7, is a promising technique. We hope that ICLR, which published the original paper, is interested publishing our paper which is a critical evaluation of the security guarantees claimed by the DI.
>
> > 2b. Moreover, there is no reason why the victim wants to use a larger number of private samples. I would imagine if the victim uses a smaller number of private samples, then correctly estimating the membership of a subset of the entirety could be sufficient to claim the ownership.
>
> A good ownership verification scheme must have a very low false positive rate. Otherwise it risks falsely accusing an independent model as a stolen model thereby undermining the trustworthiness of the ownership verification scheme. As we have shown, the ownership verification scheme is susceptible to false positives.

---

> ### Author Response · Authors · 2022-11-10
> **Response 3**
>
> > 3. Further unclear why the attacker performs adversarial training to construct a model [Use of Adversarial Training as a Defense].
>
> > 3a. The paper shows that an adversary can increase false negatives by training their model using adversarial training. This is confusing because I could not imagine a scenario where the adversary harms the accuracy of their model by running adversarial training. One of the purposes of stealing private data is to achieve state-of-the-art accuracy; then, running adversarial training contradicts the motivation.
>
> The goal of the adversary is not only to steal the model but importantly, **to remain undetected**. Hence, the adversary will use techniques that allow them to evade DI, even those that involve reasonable tradeoffs. We show that adversarial training is such a technique that an adversary can use to **completely** evade DI while incurring only a 6pp drop in accuracy. We submit that such a drop in accuracy is small enough that a defender should be concerned. To appreciate why, consider an adversary who uses the stolen model to provide a competing service at **a small fraction of what the original model owner charges** for their prediction service. Potential customers may see the model with much cheaper access with a small drop in accuracy as sufficiently cost-effective, and move their business away from the original model to the stolen model. Hence the concern.
>
> > 4. Minor:
>
> → We will address your minor comments.

---

### Author Response · Authors · 2022-11-18
**Summary of the changes**

Thanks again for your constructive feedback.

We have updated the paper based on the discussion so far. Below, we list each change we proposed (prefixed with a “→ “) to address reviewer concerns, and explain how the change was made in the paper. We have also fixed some numeric typos (no impact on the results).

**Reviewer #1 (TtqN)**

_→ This explanation is already in Section 5. We will try to clarify it better._

We expanded the discussion in Section 5 (Revealing private data).

_→ we will add experimental results with more data to the Appendix._

We added a figure for the experiments with more samples (Appendix B).

_→ We will address your minor comments._

We fixed the citation style and tweaked the phrasing of some paragraphs based on the comments.

We moved the notation table from Appendix A to Section 2.

We also moved the related work section (Section 6) to Appendix C.

**Reviewer #2 (PqLk)**

_→ We will make Theorem 2 suitable for the expectation of prediction margins + clarifications in the appendix_

We added some clarifications and details to our proofs in the Appendix A, based on the discussion.

_→ we will add the references to the paper as well as two other references we found_

_→ We will update the discussion about oblivious inference to reflect the above_

We updated the discussion in Section 5 (Revealing private data) to include additional references on oblivious computation, and to reflect the difficulty of the deployment on a large scale.

_→ We will address your minor comments._

We fixed the citation style, and clarified the phrasing based on your minor comments.

_→ In the Appendix, we will also include a figure that illustrates the results for the verification with more samples_

We added a figure for the experiments with more samples (Appendix B).

_→ We will start these experiments now._

We have added the experiments with increased noise in Appendix D.


**Reviewer #3 (ZZmg)**

_→ we will add accuracy values to Table 2._

We added model accuracies to Tables 2 (Section 3), and 3 (Section 4).

_→ We will add a plot similar to Figure 3 in the original DI paper._

We added a figure for the experiments with more samples (Appendix B).

_→ we will clarify that (a) our focus is DI in the black-box setting, and (b) the above reasoning why FPs/FNs in the black-box setting are still important findings._

We have updated the abstract and Section 1 to clarify point ‘a’ and added a discussion point ‘b’ to Section 5.

_→ We will make the subspace relation (counter-example) clearly in our paper._

_→ We will change the conclusion to "the distance between the empirical loss and the expected loss is bounded, and the bound can be very small when the model's margin is large"._

We added the clarifications in the Abstract, Introduction, and Sections 3.1 and 3.2.1.

---

### Decision · Program_Chairs · 2023-01-20

**Decision:**

Reject

**Justification For Why Not Higher Score:**

This paper in particular seemed to really suffer from lack of included code, as the reviewers were unable to replicate the results, and didn't have a reference solution from which to base their approaches on.

**Justification For Why Not Lower Score:**

 NA

**Metareview: Summary, Strengths And Weaknesses:**

Thank you for your submission to ICLR.  While there was still ultimately no consensus on this paper, there was active discussion during the the AC-reviewer meeting, which identified a number of both strengths and weaknesses of the paper.

This paper points out a number of possible issues with Dataset Inference, a technique proposed in last year's ICLR.  While the paper does not claim that DI is "broken", they do identify a number of potential issues with the approach, allowing for the possibility of either substantially more false positives *or* false negatives than was illustrated in previous papers.  Papers showing limitations of past approaches are crucial to the advance the field, and thus this paper has potentially a great deal of value to the community.  However, during the review period and the AC-reviewer discussions, the reviewers raised a number of substantial questions that raise real concerns for the work.

Specifically, several of the reviewers were quite familiar with the dataset inference problem and attempted to reproduce the results from this paper, but were unable to see the same effects as documented in this paper.  Specifically, but increasing the step size, the problem of false negatives for adversarial models and of false positives seems to be substantially alleviated.  While these points were not discussed explicitly during the rebuttal period, my inclination would be that these elements should not play a huge factor.  However, because the authors explicitly did not submit source code with the paper, issues of reproducibility are quite important, to understand if these are real effects or if these are merely due to one particular choice of hyperparameters.  Papers that explicitly point out limitation in existing work are especially crucial in this setting, and it is unclear why code couldn't be included with this work.

Given the ultimate disagreement here, I strongly recommend that the authors resubmit with code included, in order to better validate that the results presented here really do hold.

**Summary Of Ac-Reviewer Meeting:**

 The AC-reviewer meeting centered largely around the discussion on the seeming issues of this paper, from the perspective of whether the high false positive or false negative rate really held up under more rigorous evaluations of the proposed method.  The consensus is that the effects pointed out in this paper at least seemed somewhat likely to be due to choices of hyperparameters, and thus needed substantially more evidence to hold up.